# JOINTLY-LEARNED EXIT AND INFERENCE FOR A DYNAMIC NEURAL NETWORK : JEI-DNN

**Florence Regol**[*]**, Joud Chataoui**[*]**& Mark Coates**
McGill University, International Laboratory on Learning Systems (ILLS), Mila Québec AI Institute
As {florence.robert-regol, joud.chataoui}@mail.mcgill.ca
mark.coates@mcgill.ca

## ABSTRACT

Large pretrained models, coupled with fine-tuning, are slowly becoming established as the dominant architecture in machine learning. Even though these models offer impressive performance, their practical application is often limited by the prohibitive amount of resources required for *every* inference. Early-exiting dynamic neural networks (EDNN) circumvent this issue by allowing a model to make some of its predictions from intermediate layers (i.e., early-exit). Training an EDNN architecture is challenging as it consists of two intertwined components: the gating mechanism (GM) that controls early-exiting decisions and the intermediate inference modules (IMs) that perform inference from intermediate representations. As a result, most existing approaches rely on thresholding confidence metrics for the gating mechanism and strive to improve the underlying backbone network and the inference modules. Although successful, this approach has two fundamental shortcomings: 1) the GMs and the IMs are decoupled during training, leading to a train-test mismatch; and 2) the thresholding gating mechanism introduces a positive bias into the predictive probabilities, making it difficult to readily extract uncertainty information. This leads to significant performance improvements on classification datasets and enables better uncertainty characterization capabilities.

## 1 INTRODUCTION

The dominant approach to improve machine learning models is to develop larger networks that can handle every potential sample. As a result, despite very impressive performance, the resource overhead is huge (Scao et al., 2023). The push for larger model size is often driven by the need to handle a small percentage of samples that are particularly challenging to infer (Bolukbasi et al., 2017); most inferences do not need the full power of a large network to be successfully executed. Nonetheless, most traditional neural network (NN) models have a fixed processing pipeline. This means that every sample, simple or complex, is processed the same way.

To tackle this inefficiency, dynamic networks have been introduced (see (Han et al., 2022a) for a review). These models adapt the computational processing to the specific sample being processed. Early-exit dynamic networks (EEDNs) tailor their depth to the sample, allowing easy-to-infer samples to exit at shallower layers. Compared to conventional neural networks, EEDNs incorporate two additional components: 1) Intermediate Inference Modules (IMs) receive a sample's representation at their respective network depth and generate predictions; 2) Gating Mechanisms (GMs) decide which intermediate inference module should be employed to derive the final prediction.

Most EEDNs employ a simple threshold-based gating mechanism. A *threshold GM* applies thresholds to confidence metrics obtained from inference modules. To integrate more sophisticated GMs, the gating mechanism can be treated as a post-training add-on component. Given a resource budget and a set of pre-trained IMs, the optimal gating decisions are learned by taking into account both performance and inference cost. Unfortunately in both strategies, the IMs and GMs are decoupled during training; the IMs are trained on the full dataset even though they will only infer a subset of the data at inference time. This inevitably creates a mismatch between the train and test distributions.

---
[*]Equal contribution

The reliance on threshold GMs also makes it more difficult to obtain confidence levels for predictions. The information required for this determination has already been "consumed" in the gating decision; moreover, applying a threshold introduces an overconfidence bias. In a setting where a model can produce different outputs using varying levels of computational resources, confidence measures are essential. They enable end-users to make informed decisions, whether to accept a cost-efficient output or to request additional computation for a more reliable answer. Currently, state-of-the-art EEDNs do not offer confidence information.

**Contributions:** We propose a novel learning procedure for the GMs and IMs given a fixed backbone network. Our approach involves joint training so it directly avoids train-test mismatch and provides good uncertainty characterization. The method introduces a new approach for modeling the probability of exiting at a particular inference module. We optimize a loss that jointly assesses accuracy and inference cost, and formulate the minimization as a bi-level optimization task. Each level of the bi-level optimization is simpler than the overall problem, leading to more stable learning. We show empirically that the approach leads to a better overall inference/performance trade-off. The benefits are threefold: 1) we close the training gap between IMs and GMs, which leads to better performance; 2) the architecture produces reliable uncertainty characterization in the form of conformal intervals and well-calibrated predicted probabilities; and 3) our approach significantly outperforms other architecture-agnostic early-exit training procedures, while its generality facilitates integration into most state-of-the-art architectures, including those tailored for dynamic-exit.[1]

## 2 RELATED WORK

We briefly summarize the most relevant literature. Appendix 9.10 contains further discussion.

**EEDN-tailored Architectures with threshold GMs:** Much of the EEDN literature focuses on designing architectures that better lend themselves to early-exiting (EEDN-tailored architectures). BranchyNet (Teerapittayanon et al., 2016) augments the backbone network with early-exit branches and is trained using a weighted loss. Shallow-deep networks (SDNs) quantify the network confusion by comparing the outputs of different IMs (Kaya et al., 2019). Efficient end-to-end training is difficult because earlier branches influence the performance of later ones. Subsequent EEDN-tailored architectures, such as MSDNet (Huang et al., 2018) and RANet (Yang et al., 2020), alleviate this via dense connections or increased resolution. Training can be improved by reducing conflictual gradient updates using position-based rescaling (Li et al., 2019) or meta-learning (Sun et al., 2022). Reintegrating computations from earlier layers can boost performance via ensembling (Wolczyk et al., 2021; Liu et al., 2022; Passalis et al., 2020). All of these works rely on confidence score thresholds to decide whether to early-exit. The geometric ensembling of Wolczyk et al. (2021), where learnable weights adjust the contribution of each IM to the final classification decision, has connections to our exit probability modeling. Many recent advances in the EEDN field also rely on simple threshold GMs (Wang et al., 2020; Han et al., 2023b; Wang et al., 2021; Chen et al., 2023b). In general, the thresholds are treated as hyperparameters and assigned values using a validation set.

**Learnable GMs, fixed IMs:** A second class of EEDNs considers the task of learning GMs for frozen pre-trained backbones and IMs. Bolukbasi et al. (2017) train the gates sequentially using a top-down approach. In EPNet (Dai et al., 2020) the gate-training is formulated as a Markov decision process. PTEENet (Lahiany & Aperstein, 2022) employs a much simpler gate architecture. Karpikova et al. (2023) design learnable gates with a focus on confidence metrics for images. All of these models learn gating functions after having trained and frozen all of the IMs. While this two-phase procedure simplifies training, it ignores the coupling that is inherent to these two tasks.

**Addressing the train-test mismatch:** Ignoring the influence of the GMs on the IMs by training learnable gates after freezing the IMs introduces a train-test mismatch (Yu et al., 2023; Han et al., 2022b). BoostNet (Yu et al., 2023) proposes an architecture-agnostic training procedure inspired by gradient boosting to close the train-test mismatch. Deeper IMs are trained with an emphasis on samples that are incorrectly classified by earlier IMs. Han et al. (2022b) cast the problem as a meta-learning task where a *weight prediction network (WPN)* predicts the gate that will lead to the lowest classification loss and weighs the inference module losses accordingly. Although both methods are good steps towards aligning the training of the IMs with the gating mechanism, they do not fully close the train-test gap in practice. Both methods still employ threshold GMs, and the proposed specialized training of the IMs is not directly tied to which gate is ultimately selected by

---

[1]Code to reproduce our experiments is available at our Github repository

the threshold GMs. Scardapane et al. (2020) propose differentiable branching, an approach where gates and IMs are jointly learned. A straightforward modeling of the gate exit probabilities and joint optimization of all parameters results in unstable training that is slow to converge.

## 3 PROBLEM SETTING

We consider a classification problem with a training dataset $\mathcal{D} = \{\mathbf{x}_i, y_i\}_{i=1}^N$ where $\mathbf{x}_i \in \mathbb{R}^D$ denotes the input and $y_i \in \mathcal{K}$ its corresponding classification target ($\mathcal{K} = \{1, \ldots, K\}$). We are given a fixed network $NN : \mathbb{R}^D \to \mathcal{K}$ pretrained on the same task that can be decomposed into a composition of $L$ layers: $NN(\mathbf{x}) = \sigma \circ h^L \circ h^{L-1} \circ \cdots \circ h^1(\mathbf{x})$, where $\sigma$ denotes the softmax operator. Let $\mathbf{z}^l$ be the $l$-th intermediate representation of $NN$ such that $\mathbf{z}^l = h(\mathbf{z}^{l-1})$ with $\mathbf{z}^0 = \mathbf{x}$.

Our goal is to augment this fixed network (the backbone) with additional trainable components such that we can obtain a final prediction $\hat{y}_i$ with its associated predicted probability vector $\hat{\mathbf{p}}_i$ at a reduced inference cost. This setting is categorized as **intermediate classifiers-only training** (IC-only training) in the review by Laskaridis et al. (2021).

To measure the performance of our model, we consider three types of metrics: (i) performance-based (the accuracy of $\hat{y}$); (ii) uncertainty-based (the calibration error and the inefficiency score of the conformal intervals obtained from $\hat{\mathbf{p}}_i$); and (iii) efficiency-based (the inference cost to obtain $\hat{y}$).

**Inference cost:** In line with BoostedNet (Yu et al., 2023) and L2W-DEN (Han et al., 2022b) the inference cost $IC_{\hat{y}_i}$ for a single prediction $\hat{y}_i$ represents the number of multiply-add (Mul-Add) operations to obtain $\hat{y}_i$. We approximate the expected inference cost $E[IC]$ as:

$$E[IC] \approx IC = \frac{1}{N} \sum_{i=1}^N IC_{\hat{y}_i} \,. \tag{1}$$

**Uncertainty metrics:** We assess calibration of the model via the expected calibration error (ECE) of the probability of the predicted class (the maximum probability of $\hat{\mathbf{p}}_i : \hat{p}_i^{\max} = \max(\hat{\mathbf{p}}_i)$). To estimate the ECE, we follow the approach in (Guo et al., 2017). More details concerning computation of the inference cost and ECE are provided in Appendix 9.2.

To obtain conformal intervals, we use the typical conformal score $\mathbf{s}_i \triangleq \mathbf{1} - \hat{\mathbf{p}}_i$ and compute a conformal threshold $\tau^{\mathcal{V},\alpha}$ from a validation set $\mathcal{V}$ so that we can guarantee a **coverage** of $1 - \alpha$. (See Appendix 9.3 for details). A conformal interval $\hat{\mathcal{C}}_i$ for a sample $\mathbf{x}_i$ is then constructed by thresholding the conformal score $\mathbf{s}_i : \mathcal{C}_i = \{k; \mathbf{s}_i^{(k)} < \tau^{\mathcal{V},\alpha}\}$, where $\mathbf{s}^{(k)}$ denotes the $k$-th element of $\mathbf{s}$.

We measure the performance of our predicted conformal intervals $\hat{\mathcal{C}}_i$ with two metrics: **empirical coverage** $1 - \hat{\alpha}$, which computes the empirical probability that the ground truth is contained in the interval, and **inefficiency** $|\bar{\mathcal{C}}|$, which is the average cardinality of the set intervals:

$$|\bar{\mathcal{C}}| \triangleq \frac{1}{N} \sum_{i=1}^N |\hat{\mathcal{C}}_i| \,, \quad 1 - \hat{\alpha} \triangleq \frac{1}{N} \sum_{i=1}^N \mathbb{1}[y_i \in \hat{\mathcal{C}}_i] \,. \tag{2}$$

A well-behaved conformal prediction model has small inefficiency $|\bar{\mathcal{C}}|$ (intervals are small on average) while maintaining an empirical coverage close to the desired one: $1 - \alpha \approx 1 - \hat{\alpha}$.

## 4 METHODOLOGY: JEI-DNN

We introduce our method named: Jointly-Learned Exit and Inference for a Dynamic Neural Network (JEI-DNN). Our model consists of augmenting a pretrained and fixed network by attaching $L$ classifiers and $L$ gates to the intermediate layers of the network architecture. We refer to each of these classifiers as an inference module (IM). The $l^{th}$ IM, $f_{\theta_l}^l(\cdot)$, uses the $l^{th}$ representation $\mathbf{z}_i^l$ of the input $\mathbf{x}_i$ to model class probabilities $\hat{\mathbf{p}}_\theta^l(\mathbf{x}_i) = f_\theta^l(\mathbf{z}_i^l)$ and to obtain a prediction $\hat{y}_i^l = \arg\max_k \hat{\mathbf{p}}_\theta^{l,(k)}(\mathbf{x}_i)$. We set the last IM to be the original classifier of the pretrained network: $f^L = h^L$. The output of the $l^{th}$ gate, $e^l \in \{0, 1\}$, is modelled by a random variable $E^l|X$ that specifies whether the $l$-th exit gate is open. If a sample reaches the $l$-th gate and it is open, then it exits via the $l^{th}$ IM. Figure 1 depicts the JEI-DNN learning procedure and highlights the bi-level optimization used for joint training.

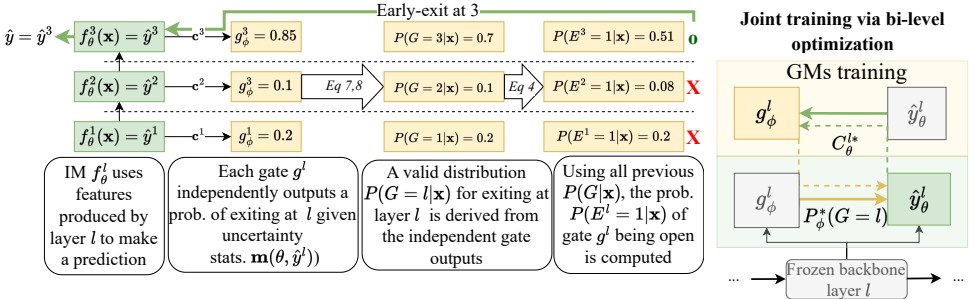

Figure 1: **Left:** Modelling of GM and how early exiting is achieved. **Right** Illustration of the mutual influence of GM and IM.

The assumption is that some samples are "easy enough" to be classified by the early classifiers $f_\theta^l(\mathbf{z}^l)$. Since the last classifier should be able to classify the entire dataset, the last gate is always set to one, i.e., $e^L = 1$. Denoting the indicator function by $\mathbb{I}[\cdot]$, the output of our model $\hat{y}$ is:

$$\hat{y}(\mathbf{x}) = \hat{y}^l(\mathbf{x}) \text{ where } l = \underset{l \in [L]}{\arg\min} \, \mathbb{I}[e^l(\mathbf{x}) = 1], \quad e^l(\mathbf{x}) \sim p(E^l|X = \mathbf{x}). \tag{3}$$

We can view each IM $f_\theta^l$ as solving a subtask over a subset of data $\mathcal{D}^l \subset \mathcal{D}$, where the partitioning is determined by the gates, and $\mathcal{D}^j \cap \mathcal{D}^k = \varnothing$ for $j \neq k$ and $\cup_{l \in [L]} \mathcal{D}^l = \mathcal{D}$. Since the $E^l$ are random, we can also define the distribution $P(G)$ of exiting at a gate $G \in [L]$. This equals the probability that the sample does not exit at any previous gate multiplied by the probability that it exits at gate $l$:

$$P(G = l|X = \mathbf{x}) = P(E^l|X = \mathbf{x}) \prod_{j=1}^{l-1} \Big( 1 - P(E^j|X = \mathbf{x}) \Big). \tag{4}$$

The first $l$ network layers and the first $l$ IMs and gates must be evaluated in order to exit at the $l^{th}$ layer. Assuming fixed-size input samples, the inference cost $IC_{\hat{y}_i^l}$ associated with obtaining a prediction from the $l^{th}$ IM can be approximated as a constant $IC^l$ that only depends on $l$. For all predictions $\hat{y}_i^l$ obtained at level $l$ we then have $IC_{\hat{y}_i^l} = IC^l$. Appendix 9.2 provides more detail about how $IC^l$ is computed. During training we use a normalized version $IC_{\text{norm}}^l = \frac{IC^l}{IC^L}$ that represents the relative cost incurred by exiting at $l$ compared to going through the full network (i.e., all $L$ layers).

We can then formulate our loss as the expectation over the exited gates $G|X$ and the data distribution $X, Y$ of the traditional cross entropy loss $\mathcal{L}^{CE}(Y, \hat{\mathbf{p}}_\theta^{G|X}(X))$ added to the associated inference cost $IC^{G|X}$ scaled by a parameter $\lambda$ that controls the importance of the inference cost:

$$\mathcal{L} = \mathbb{E}_{Y,X} \mathbb{E}_{G|X} \big[ \mathcal{L}^{CE}(Y, \hat{\mathbf{p}}_\theta^{G|X}(X)) + \lambda IC_{\text{norm}}^{G|X} \big], \tag{5}$$

$$\mathcal{L} \approx \frac{1}{N} \sum_{i=1}^{N} \Big( \sum_{l=1}^{L} (\mathcal{L}^{CE}(y_i, \hat{\mathbf{p}}_\theta^l(\mathbf{x}_i)) + \lambda IC_{\text{norm}}^l) P(G = l|\mathbf{x}_i) \Big). \tag{6}$$

Our goal is to simultaneously learn the parameters $\theta$ of the IMs and choose $P(G = l|\mathbf{x}_i)$ in order to minimize the expected loss as shown in Figure 1. We target $P(G = l|\mathbf{x}_i)$ rather than $P(E^j|\mathbf{x}_i)$ for $j = 1, \ldots, l$ because directly learning the $P(E^j)$ leads to optimization issues. The product of probabilities across multiple layers can quickly vanish or saturate to 1.

**Modeling $P(G|\mathbf{x}_i)$.** In practice, we only need to evaluate $P(G = l|\mathbf{x}_i)$ if the dynamic evaluation has reached that point, i.e., everything up to $f_\theta^l(\mathbf{z}_i^l)$ has been evaluated and is accessible. Therefore, our parameterization of $P(G = l|\mathbf{x}_i)$ can include any intermediate values calculated by the base architecture, the inference modules, and the gates up to and including layer $l$. We denote this aggregation of information as $\mathbf{c}_i^{\leq l}$.

This however also implies that we cannot directly model the multiclass distribution $P(G|\mathbf{x}_i)$ with the traditional softmax approach since it would require $f_\theta^l(\mathbf{z}_i^l)$ to be evaluated for all $l$. This would

defeat the purpose of early-exiting. Instead, we model each $P(G = l|\mathbf{x}_i)$ sequentially, starting from the first layer $l = 1$, with a learnable variable $g_\phi^l(\mathbf{c}_i^{\leq l}) \in [0, 1]$. We set $P(G = l|\mathbf{x}_i)$ to the minimum of the learnable variable $g_\phi^l(\mathbf{c}_i^{\leq l}) \in [0, 1]$ and $1 - \sum_{j=1}^{l-1} g_\phi^j(\mathbf{c}_i^{\leq j})$. Since the model must exit at the final gate if it has not already done so, we have $g^L(\mathbf{c}_i^{\leq L}) = 1$. Thus:

$$P_\phi(G = 1|\mathbf{x}_i) = g_\phi^1(\mathbf{c}_i^{\leq 1}), \tag{7}$$

$$P_\phi(G = l|\mathbf{x}_i) = \min(g_\phi^l(\mathbf{c}_i^{\leq l}), 1 - \sum_{j=1}^{l-1} g_\phi^j(\mathbf{c}_i^{\leq j})) \quad \text{for} \quad l = 2, \dots, L. \tag{8}$$

The $\min$ operator in Eq. (8) ensures that the $\{P_\phi(G = l|\mathbf{x}_i); l = 1, \dots, L\}$ form a valid probability distribution. The loss is then a proper approximation of the expectation in Eq. (5). It also guarantees that $P(E^l|X = \mathbf{x})$, which is calculated using (4) and governs the decision to exit, lies in $[0, 1]$.

This formulation leads to a much more stable model for the gate probabilities compared to a parameterization through $P(E^l|\mathbf{x})$. A deviation in one of the $g_\phi^l$s has an additive rather than multiplicative impact on subsequent $P_\phi(G = l|\mathbf{x}_i)$s. The left diagram in Figure 1 depicts the modeling of $P(G|\mathbf{x}_i)$. Our approximated loss from equation 6 is then:

$$\mathcal{L} \approx \frac{1}{N} \sum_{i=1}^N \sum_{l=1}^L (\mathcal{L}^{CE}(y_i, \hat{\mathbf{p}}_\theta^l(\mathbf{x}_i)) + \lambda IC_{\text{norm}}^l) \min \left( g_\phi^l(\mathbf{c}_i^{\leq l}), 1 - \sum_{j=1}^{l-1} g_\phi^j(\mathbf{c}_i^{\leq j}) \right). \tag{9}$$

Any parameterization of the IMs $f_\theta^l$ and gates function $g_\phi^l(\mathbf{c}_i^{\leq j})$ can be adopted. However, the modules must be small compared to the size of one layer of the backbone model to prevent them from substantially increasing the computational cost of the entire layer during inference. In our experiments, $f_\theta^l$ and $g_\phi^l(\mathbf{c}_i^{\leq j})$ are parameterized as simple one-layer neural networks and uncertainty metrics are extracted from $\mathbf{c}_i^{\leq j}$ to serve as input to $g_\phi^l(\cdot)$. A detailed description of our modules is included in Section 4. Appendix 9.2 includes the inference cost of the gates and IMs to demonstrate that the added computation is minimal when compared to the inference cost of a single layer. (The total added computation amounts to less than 0.01% of the Mul-Adds of the backbone.)

**Uncertainty prediction** : Next we describe how we use the predicted probabilities $\hat{\mathbf{p}}_\theta^l(\mathbf{x}_i)$ to form conformal intervals. The conformal intervals are derived from a conformal threshold $\tau^{\mathcal{V},\alpha}$ that is computed using a validation set $\mathcal{V}$. However, since we can view the IMs as solving different subtasks over different subsets of the data $\mathcal{D}^l$, we should compute a different conformal threshold for each subtask. Hence, we form one validation set per gate, $\mathcal{V}^l$, where $\mathcal{V}^l = \{i; l \sim P_\phi(G|\mathbf{x}_i), i \in \mathcal{V}\}$, and compute a conformal threshold from that subset: $\tau^{\mathcal{V}^l,\alpha}$. If there are too few samples in the validation set associated with gate $l$, we set $\tau^{\mathcal{V}^l,\alpha}$ to a general threshold $\tau^{\mathcal{V},\alpha}$ computed using the entire validation set. The conformal intervals are then constructed as follows:

$$\mathcal{C}_i = \{k; 1 - \hat{\mathbf{p}}_\theta^{l,(k)}(\mathbf{x}_i) < \tau^{\mathcal{V}^l,\alpha}, l \sim P(G|\mathbf{x}_i)\} \tag{10}$$

There are many other ways to construct the sets to compute the threshold in the EDNN setting. In Appendix 9.3, we show that a variety of methods produce similar results.

We present candidate designs for the gates and IMs for concreteness, but note that other choices are possible within the JEI-DNN framework.

**Gate design** $g_\phi^l(\mathbf{c}_i^{\leq l})$**:** To ensure a lightweight design, we construct a small number of features by computing uncertainty statistics from the output $\hat{\mathbf{p}}^l$ of the $l$-th IM, $f_\theta^l(\mathbf{x}_i)$. Hence the $l$-th gate can be represented as $g_\phi^l(\mathbf{c}_i^{\leq l}) = g_\phi^l(m(\theta, \mathbf{x}_i))$. We choose $m(\theta, \mathbf{x}_i) = [\hat{p}_i^{l,\max}(\mathbf{x}_i), h^l(\mathbf{x}_i), h_{pow}^l(\mathbf{x}_i), mar^l(\mathbf{x}_i)]^T$ where $\hat{p}_i^{l,\max}(\mathbf{x}_i)$ is maximum predicted probability, $h^l(\mathbf{x}_i)$ is the entropy, $h_{pow}^l(\mathbf{x}_i)$ is the entropy scaled by a temperature, and $mar^l(\mathbf{x}_i)$ is the difference between the two most confident predictions. For more detail see Appendix 9.9.

**IM design** $f_\theta^l(\mathbf{z}_i^{\leq l})$**:** The IMs $f_\theta^l(\mathbf{z}_i^{\leq l})$ all have the same architecture. We reduce cost by limiting their input size and number of layers. The input to $f_\theta^l$ is only $\mathbf{z}_i^l$ and the IM is a single layer NN. The lightweight design allows us to insert exit branches at every layer. This gives our model greater exit flexibility, in contrast with many existing approaches (Chiang et al., 2021; Lahiany & Aperstein, 2022; Li et al., 2023; Ilhan et al., 2023), which typically use only a few exit branches.

## 4.1 Optimization

Directly optimizing the loss in equation 9 is challenging because of the $\min$ operator:

$$\phi^*, \theta^* = \arg\min_{\phi,\theta} \frac{1}{N} \sum_{i=1}^{N} \sum_{l=1}^{L} (\mathcal{L}^{CE}(y_i, \hat{\mathbf{p}}_\theta^l(\mathbf{x}_i)) + \lambda I C_{\text{norm}}^l) \min\left(g_\phi^l(\mathbf{c}_i^{\leq l}), 1 - \sum_{j=1}^{l-1} g_\phi^j(\mathbf{c}_i^{\leq j})\right). \quad (11)$$

However, there is a distinction between the parameters $\theta$ and $\phi$ which makes this loss a good candidate for a bi-level optimization formulation. (See (Chen et al., 2023a) for a survey.) Hence, using $C_{\theta(\phi)}^l = \mathcal{L}^{CE}(y_i, \hat{\mathbf{p}}_\theta^l(\mathbf{x}_i)) + \lambda I C_{\text{norm}}^l$, we can rewrite equation 11 as follows:

$$\phi^* = \arg\min_\phi \mathcal{L}^{out} \triangleq \arg\min_\phi \frac{1}{N} \sum_{i=1}^{N} \sum_{l=1}^{L} C_{\theta^*(\phi)}^l P_\phi(G = l|\mathbf{x}_i), \quad (12)$$

$$s.t. \quad \theta^*(\phi) = \arg\min_\theta \mathcal{L}^{in} \triangleq \arg\min_\theta \frac{1}{N} \sum_{i=1}^{N} \sum_{l=1}^{L} (\mathcal{L}^{CE}(y_i, \hat{\mathbf{p}}_\theta^l(\mathbf{x}_i) + \lambda I C_{\text{norm}}^l) P_\phi(G = l|\mathbf{x}_i). \quad (13)$$

Since the gate probabilities $P_\phi(G = l|\mathbf{x}_i)$ take as input the aggregation of any intermediate values that were previously calculated $\mathbf{c}_i^{\leq l}$, it could be dependent on the $\theta$. To make that dependence explicit in the calculation of the gradient of $\theta$, we denote $P_\phi(G = l|\mathbf{x}_i) = G_\phi(m(\theta, \mathbf{x}_i))$. The derivative of the loss with respect to the two different set of parameters is given by:

$$\frac{\partial \mathcal{L}^{in}}{\partial \theta} = \frac{1}{N} \sum_{i=1}^{N} \sum_{l=1}^{L} \frac{\partial \mathcal{L}^{CE}(y_i, \hat{\mathbf{p}}_\theta^l(\mathbf{x}_i))}{\partial \theta} P_\phi(G = l|\mathbf{x}_i) + \frac{\partial G_\phi(m(\theta, \mathbf{x}_i))}{\partial \theta} C_{\theta(\phi)}^l, \quad (14)$$

$$\frac{\partial \mathcal{L}^{out}}{\partial \phi} = \frac{1}{N} \sum_{i=1}^{N} \sum_{l=1}^{L} \frac{\partial P_\phi(G = l|\mathbf{x}_i)}{\partial \phi} C_{\theta^*(\phi)}^l = \frac{1}{N} \sum_{i=1}^{N} \sum_{l=1}^{L} \frac{\partial \min\left(g_\phi^l(\mathbf{c}_i^{\leq l}), 1 - \sum_{j=1}^{l-1} g_\phi^j(\mathbf{c}_i^{\leq j})\right)}{\partial \phi} C_{\theta^*(\phi)}^l. \quad (15)$$

Eq. (15) is undefined when the $\min$ operator terms are equal. We address this issue below when describing how to optimize the gates.

**Optimizing the IMs:** By inspecting equation 14, we can recognize that the left term is the same gradient that is encountered for a straightforward weighted cross-entropy loss. Interestingly, our principled approach leads to an objective with a term that is similar to the one proposed by Han et al. (2022b) to address the train-test mismatch issue. In our case, the weights emerge directly from our proposed gating mechanism: $P_\phi(G = l|\mathbf{x}_i)$. The right term is driven by the impact of the $\theta$ parameter on the gates: $\frac{\partial G_\phi(m(\theta, \mathbf{x}_i))}{\partial \theta}$. In practice, we observe that ignoring the right term of the gradient does not impact performance and leads to faster convergence, as we show in Appendix 9.7.

**Optimizing the gates** The second derivative in equation 15 is more challenging, making direct optimization undesirable. Instead, we construct surrogate binary classification tasks to train the $g_\phi^l$. For a given sample $\mathbf{x}_i$ we construct binary targets for $g_\phi^1(\mathbf{c}_i^{\leq 1}), ..., g_\phi^{L-1}(\mathbf{c}_i^{\leq L-1})$ by evaluating the relative cost of each gate $C_{\theta^*(\phi)}^l$ and determining which of the $L$ gates has the lowest cost, denoted as $l^* = \arg\min_l C_{\theta^*(\phi)}^l$. This is the gate at which $\mathbf{x}_i$ should exit. Hence, we set the binary target of gate $l^*$ to 1, and the binary targets for all of the preceding gates to zero. As for the subsequent gates, since the sample is supposed to exit at $l^*$, we assume that it should also exit at any later gate, so we set the binary targets of all subsequent gates to 1 as well. Hence, the targets of our binary tasks $t_i^1, ..., t_i^L$ for our gates $g_\phi^1(\mathbf{c}_i^{\leq 1}), ..., g_\phi^{L-1}(\mathbf{c}_i^{\leq L-1})$ are given by $t_i^j = 0$ for $j < l^*$ and $t_i^j = 1$ for $j \geq l^*$.

The connection between those surrogate tasks and the initial objective from equation 12 can be established by showing that they can share the same solution under some assumptions. We demonstrated this is Appendix 9.4. Hence, instead of following the gradient from equation 15, we approximate it by the gradient of the surrogate loss:

$$\frac{\partial \mathcal{L}^{out}}{\partial \phi} \approx \frac{1}{N} \sum_{i=1}^{N} \sum_{l=1}^{L} \frac{\partial \mathcal{L}^{CE}(t_i^l, g_\phi^l(\mathbf{c}_i^{\leq l}))}{\partial \phi}. \quad (16)$$

This amounts to summing the gradients of the losses of $L$ independent binary classification tasks. If the tasks are highly imbalanced, we compute the class imbalanced ratio on a validation set at each epoch and use weighted class training. Following the bi-level optimization procedure, training is carried out by alternating between optimizing using the gradients in equation 16 and equation 14. In practice, we start by training all IMs on the full dataset in a warmup stage as described in Section 5. Algorithm 1 in Appendix 9.5 provides a detailed exposition of the entire algorithm.

## 5 EXPERIMENTS

We use the vision transformers T2T-ViT-7 and T2T-ViT-14 (Yuan et al., 2021) pretrained on the ImageNet dataset (Deng et al., 2009) which we then transfer-learn to the datasets: CIFAR10, CIFAR100, CIFAR100-LT (Krizhevsky, 2009) and SVHN (Netzer et al., 2011). For dataset details, see Appendix 9.1.1. For transfer-learning, we use the procedure from (Yuan et al., 2021). We provide checkpoints for all these models in our code. Our backbone $NN$ for each dataset is the frozen transfer-learned model. We augment the backbone with gates and intermediate IMs at every layer. We generate results by varying the inference cost parameter $\lambda$ over the range 0.01 to 10. Rather than sample $E^l \sim p(E^l|\mathbf{x})$ to decide whether to exit, we use the deterministic decision $P(E^l) > 0.5$. Appendix 9.1.1 contains further details concerning hyperparameters and experimental procedure.

**Training procedure**: Our training procedure consists of two phases: (i) warm-up; and (ii) bi-level optimization. In the warm-up phase the first $L-1$ IMs are trained in parallel on all samples for a fixed number of warm-up epochs. This ensures that intermediate IMs are performing reasonably well when we start training the gates. During the bi-level optimization, we alternate between optimizing the gate parameters $\phi$ and the IM parameters $\theta$. See Appendix 9.1.4 for more detail.

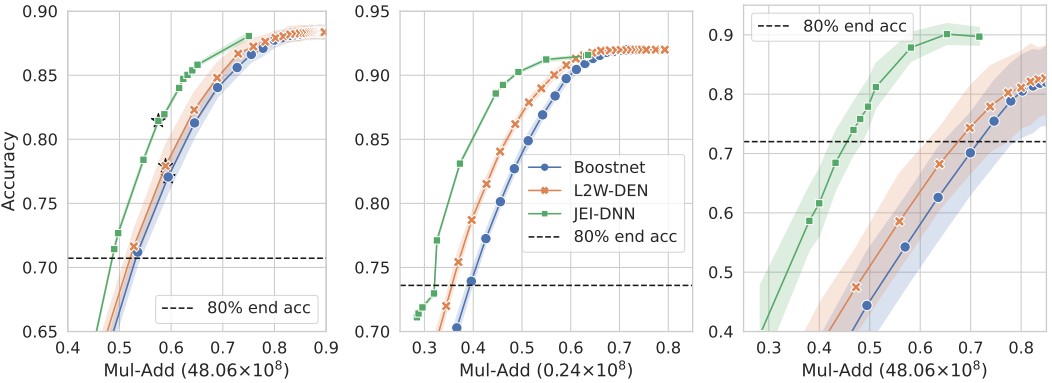

Figure 2: Accuracy vs Mul-Add of: **Left** CIFAR100 (t2t-14); **Middle** SVHN (t2t-7); and **Right** CIFAR100LT (t2t-14). The x-axes are scaled by the full model inference cost, Mul-Add ($IC^L$).

**Baselines**: We compare our proposal with the following baselines:

- **BoostedNet** (Yu et al., 2023) and **L2W-DEN** (Han et al., 2022b) are state-of-the-art benchmarks with architecture-agnostic training procedures for general-purpose networks.
- **MSDNet** (Huang et al., 2018) and **RANet** (Yang et al., 2020) are EEDN-tailored architectures. We also compare with the improved training procedures in (Yu et al., 2023) and (Han et al., 2022b).

To obtain the uncertainty metrics, we calibrate the predicted $\hat{\mathbf{p}}$ by first applying the post-calibration algorithm of temperature scaling from (Guo et al., 2017) on the output logits using a validation set. Since all baselines also require a validation set to select gating thresholds, we split the validation set into two $\mathcal{V} = \mathcal{V}^1 \cup \mathcal{V}^2$; $\mathcal{V}^1$ is used to compute the gating thresholds and $\mathcal{V}^2$ is used for hyperparameter tuning, early stopping, setting the conformal threshold, and calibrating the output.

### 5.1 RESULTS

**Observation 1:** *Joint Training of the GMs and the IMs closes the train-test mismatch which leads to significant performance improvement.*

Figure 2 depicts performance versus inference cost curves. Appendix 9.8 presents results on additional datasets and architectures. Focusing on the region exceeding 80% of the end accuracy of the total network, our suggested approach significantly outperforms the baselines for all datasets. Outside of that range, it is on par or better. Moreover, the 95% confidence intervals around the mean of the accuracy of our proposed approach (shaded areas) are significantly tighter. The CIs of all methods are wider for CIFAR100-LT. The imbalance leads to noisier prediction, which makes it particularly challenging for the baselines. To provide further insight into why our model outperforms the baselines, we analyze the results in the subsequent sections. Appendix 9.6 provides an ablation study which demonstrates the value of learnable gates and joint training.

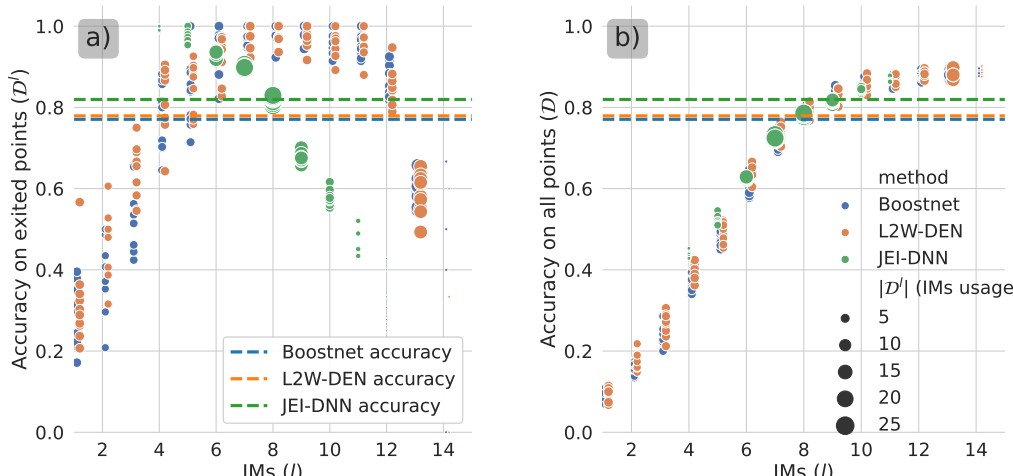

Figure 3: Decomposition of the contributions of the IMs to the final accuracies (depicted by dotted lines) for CIFAR100. The operational point is marked by a star in the left panel of Figure 2. **a)** Accuracy of each IM, $f_\theta^l$, evaluated only on their exited samples ($\mathcal{D}^l$). **b)** Accuracy of each IM, $f_\theta^l$, on the full dataset $\mathcal{D}$. The size of a circle is proportional to the number of samples exited for a trial.

**Observation 2:** *Training the GMs leads to a better gate selection.*

Because our GMs are jointly trained with the IMs and take into account the cost of inference, the GMs can avoid poorly performing gates or gates that provide only marginal performance improvement. Figure 3 shows which gates are selected for exit by our proposed method (JEI-DNN) and the baselines. We see that JEI-DNN avoids using early IMs altogether and focuses on IMs 5 to 10. The joint training approach thus effectively addresses the EEDN placement task. Rather than employing fewer more capable, complex GMs and IMs selecting where to place them, as in (Li et al., 2023), our approach introduces very simple, lightweight GMs and IMs, and learns to focus on the IMs that can perform satisfactorily for a given inference budget. In contrast, the threshold GMs employed by the baselines lead to more evenly distributed exits. Since the threshold GMs make decisions by thresholding confidence metrics derived from predicted probabilities, it is more difficult to avoid exiting at early poorly performing IMs (these IMs are overly-confident for some samples).

Moreover, the thresholding approach relies on the assumption that IMs are well calibrated, which is not guaranteed (Guo et al., 2017; Minderer et al., 2021). Experimentally, we verify that it is not the case and that calibration is positively correlated with IM depth (see Figure 4). Consequently, calibration is also correlated with accuracy, which is in line with the findings of Minderer et al. (2021)). A poor calibration leads to inaccurate predicted probabilities, which introduces noise in the inputs to the threshold GM. As a result, the exit selection is also more variable, leading to a less predictable accuracy-IC trade-off (as exhibited by the much wider CIs in Figure 2).

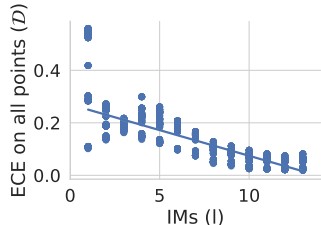

Figure 4: ECE of the IMs averaged over all baselines on all datasets with t2t-7.

**Observation 3:** *Better gate selection concentrates training on better IMs.*

Figure 3(b) highlights that the improvement of our model does not come from IMs that are superior over all of the training data. Baseline and JEI-DNN IMs perform similarly. The improvement in IC-accuracy trade-off is explained by *the trend* of IM accuracies in conjunction with IM usage. JEI-DNN exits very few samples at IMs 1-4 and 11-14. The early IMs are too inaccurate. The accuracy gain achieved by postponing exit to late IMs (11-14) is not justified by the increased inference cost. The GMs must be able to accurately direct easier samples to earlier IMs. JEI-DNN clearly achieves this. For IMs 6-8, where most samples exit, the accuracy is higher over exiting samples (Fig. 3(a)) than all samples (Fig. 3(b)). By contrast, for gates 9-10, where only a few challenging samples exit, the exit accuracy is lower. For JEI-DNN, the most challenging samples exit at gates 9-11; for the baselines, they exit at gate 13. Despite having access to less informative features, JEI-DNN achieves a better average accuracy for these challenging samples at considerably lower inference cost.

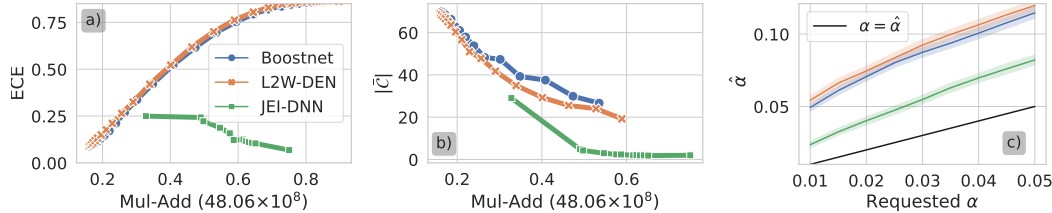

Figure 5: Uncertainty metrics on CIFAR-100 (t2t-14). **a)** ECE vs Mul-Add. **b)** Inefficiency vs Mul-Add for an empirical coverage bounded by $1 - \hat{\alpha} \geq 95\%$ on CIFAR-100 (t2t-14). **c)** Average empirical $\hat{\alpha}$ vs requested $\alpha$. Appendix 9.8 presents results on other datasets and architectures.

**Observation 4:** *Learnable GMs leads to better uncertainty characterization capability*

Fig. 5**(a)** highlights that even with post-calibration, the baselines' predicted probabilities remain extremely poorly calibrated. JEI-DNN's calibration is much better and the calibration error correlates negatively with accuracy. This is desirable; a higher inference cost should lead to better accuracy and better calibration. When the baselines provide more accuracy, the calibration error is much worse. This disparity has two causes: (i) JEI-DNN refrains from using the early, worst-calibrated gates (Fig. 4); (ii) the baselines must increase the GM thresholds to achieve higher accuracy, but this leads to a bias — samples only exit if the predictions are very confident. We now consider conformal intervals. These should circumvent the threshold GM bias as they are derived via a validation set and have probabilistic guarantees. Fig. 5**(b)** shows that JEI-DNN offers significantly tighter conformal intervals $|\bar{\mathcal{C}}|$ for a given constraint on empirical coverage $\hat{\alpha}$. The empirical coverage values for JEI-DNN are also closer to the requested coverages $\alpha$ (see Fig. 5**(c)**).

**Observation 5:** *JEI-DNN can enhance SOTA dynamic architectures*

We examine whether JEI-DNN can enhance a state-of-the-art EEDN-tailored architecture. We select the Dynamic Perceiver Han et al. (2023b) with MobileNetV3 (Howard et al., 2019) as the backbone. The combination (DynPerc+JEI-DNN) includes learnable gates. We train for 310 epochs. During the last 10 epochs for DynPerc+JEI-DNN, JEI-DNN is used to jointly train gates and IMs. Fig. 6 shows that this leads to significant accuracy improvements for both CIFAR100 and ImageNet.

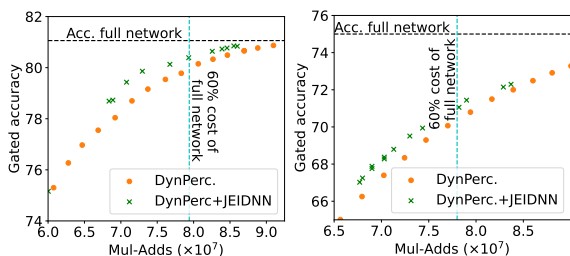

Figure 6: DynPerceiver+JEIDNN vs. DynPerceiver on CIFAR100 **(left)** and ImageNet **(right)** (MobileNetV3).

## 6 LIMITATIONS

Our proposal outperforms the baselines in terms of the accuracy-IC trade-off and uncertainty characterization, but it has some drawbacks. Since we must specify the inference cost parameter $\lambda$, we only obtain one accuracy/IC point per training. To obtain different values, we need to retrain the model with a changed $\lambda$ value. By contrast, the baseline methods that employ threshold GMs can modify the inference time after deployment by changing the threshold. Second, we cannot directly target a specific accuracy or inference cost. In practice, if there is a constraint on one of the metrics, we either must train multiple models, adjusting $\lambda$ until the constraint is met, or use an ad-hoc approach by applying different thresholds to the $g_\phi^l$ values. In summary, if the desired trade-off between inference cost and accuracy is likely to vary over time, and it is acceptable to sacrifice some accuracy, then the approach taken by the baselines would be preferable.

## 7 CONCLUSION

We have introduced a novel early-exit dynamic neural network learning procedure, JEI-DNN, that is compatible with off-the-shelf backbone architectures. By augmenting a backbone with lightweight, trainable gates and inference modules that are jointly optimized, we close the train-test gap, leading to better performance of inference modules. The approach can improve state-of-the-art EEDN-tailored architectures and provides significantly better uncertainty characterization. The inference modules are better calibrated and tighter conformal intervals are obtained for a desired coverage.

## 8 Acknowledgments

Cette recherche a été financée par le Conseil de recherches en sciences naturelles et en génie du Canada (CRSNG), [numéro de référence 260250].
We acknowledge the support of the Natural Sciences and Engineering Research Council of Canada (NSERC), [funding reference number 260250].
We also thank Boris Oreshkin for kindly providing additional computational resources.

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

# 9 APPENDIX

## 9.1 EXPERIMENT DETAILS

### 9.1.1 DATASETS

The CIFAR10 and CIFAR100 (Krizhevsky, 2009) datasets both consist of 60,000 $32 \times 32$ coloured images. CIFAR10 has 10 classes and CIFAR100 has 100 classes. We follow a 75%-8.3%-16.6% train-validation-test split. The validation set is used for hyperparameter tuning and early-stopping. ImageNet (Deng et al., 2009) consists of 1.2 million training images spanning 1,000 classes. We reserved 50,000 of these images to be used as validation set and used another 50,000 images as a test set. Following the data transform of Han et al. (2023b), the images are cropped to be of size $224 \times 224$. We also use the cropped-digit SVHN (Netzer et al., 2011) dataset which consists of 99,289 $32 \times 32$ coloured images of house numbers where the target is the digit in the center. We use a 68.6%-5%-26.2% train-validation-test split. The CIFAR100-LT (long-tail) dataset consists of the regular CIFAR100 images but with imbalanced classes. We use an imbalance factor $if = 100 = \frac{|c_{\text{most common}}|}{|c_{\text{least common}}|}$. We follow the pre-processing procedure from (Yuan et al., 2021). For all datasets, we use a training batch size of 64. We present 95% confidence intervals on the means of our results. These are generated using a bootstrap procedure over 10 trials derived by partitioning the test set into 10 subsets.

### 9.1.2 EXPERIMENT PARAMETERS

We use the Adam optimizer with a learning rate of $0.01$ with a weight decay of $5e{-}4$. We use a batch size of 64 for CIFAR10, CIFAR100 and CIFAR100LT, and a batch size of 256 for SVHN. We train until convergence using early stopping with a maximum of 15 epochs ($E$) for CIFAR10 and SVHN, 20 for CIFAR100LT and 30 for CIFAR100LT. For the number of warmup epochs $WE$, we perform a hyperparameter search over the range $\{1, \ldots, E/2\}$.

### 9.1.3 TRAINING TIME AND MEMORY USAGE

We report the average training time on a GPUx machine and number of parameters used for both architectures in Table 1. Overall, our approach takes a little longer to train and has a negligible number of additional parameters. Once trained, the baselines do have the advantage that heuristic adjustment of the gating threshold can achieve different IC/accuracy trade-offs.

Table 1: Memory usage and training time comparison. *We note that training time is to obtain a single IC/accuracy trade-off model (one point in our curve).

|  | JEI-DNN | Boostnet | L2W-DEN |
|---|---|---|---|
| **T2T-7 (CIFAR10)** | | | |
| time* (min) | 15 | 13 | 13 |
| num. parameters | 15.45K | 15.42K | 15.42K |
| **T2T-14 (CIFAR100)** | | | |
| time* (min) | 37 | 22 | 24 |
| num. parameters | 500.56K | 500.50K | 500.50K |

### 9.1.4 WARM-UP LOSS

We now provide a more thorough exposition of the warm-up phase which precedes the bi-level optimization phase. As mentioned, during warm-up we train all IMs in parallel on all samples. This is done to ensure that IMs have an acceptable accuracy and can potentially be chosen as exit points when we start training the gates. Earlier IMs have lower representational capacity and thus require more training. To that end, we increase the learning rate of earlier IMs using a simple position-based scaling. The warm-up loss for the $l^{th}$ IM can thus be written as:

$$\mathcal{L}_{warmup}^{l} = \frac{1}{N} \sum_{i=1}^{N} (L - l) lr_0 \mathcal{L}^{CE}(y_i, \hat{\mathbf{p}}_{\theta^*}^{l}(\mathbf{x}_i)), \tag{17}$$

where $lr_0$ is set to 0.01 as a hyperparameter.

### 9.2 INFERENCE COST AND EXPECTED CALIBRATION ERROR (ECE)

**Inference cost:** Following BoostNet (Yu et al., 2023) and L2W-DEN (Han et al., 2022b), we measure the inference cost in terms of multiply-add operations (Mul-Add). We use Meta's FAIR fvcore library for computing Mul-Adds. We obtain results that match the Mul-Adds reported in Yuan et al. (2021) for the T2T-ViT architecture. Since our training procedure directly relies on the inference cost (see equation 6) we measure the number of Mul-Adds at each exit before training, accounting for the layer of the backbone network, denoted $h^l$, the IM $f_{\theta_l}^l$ and the gate $g_{\phi_l}^l$. This value is treated as a function of $l$ that is constant across all samples exiting at layer $l$. That is, we ignore any possible decrease in Mul-Adds that might arise from sparsity in certain samples and thus use a worst-case scenario value that depends only on the network architecture and the input size. We can thus define the inference cost incurred for exiting at layer $l$:

$$\text{IC}^l = \sum_{k=1}^{l} \text{Mul-Add}(h^k) + \text{Mul-Add}(f_{\theta_k}^k) + \text{Mul-Add}(g_{\phi_k}^k) \tag{18}$$

Tables 2 and 3 show the number of Mul-Adds at each exit of the base T2T-ViT-7 architecture on CIFAR10 and the base T2T-ViT-14 architecture on CIFAR100 respectively as well as the added Mul-Adds incurred from augmenting the backbone network with exit branches (IMs and gates) at every layer. We note that overall, this leads to an almost insignificant inference cost increase of less than 0.003% for the former and 0.01% for the latter. As a result of our very lightweight gate design, optimal branch placement is unnecessary.

Table 2: Comparison of the number of Mul-Adds per layer between the base T2T-ViT-7 backbone and JEI-DNN. The middle rows show the net increase due to augmenting with IMs and gates.

| | $IC^1$ | $IC^2$ | $IC^3$ | $IC^4$ | $IC^5$ | $IC^6$ | $IC^7$ |
|---|---|---|---|---|---|---|---|
| T2T ViT 7 | 414.3M | 538.2M | 662.1M | 786M | 909.9M | 1034M | 1158M |
| Cum. ↑ from IMs | +2570 | +5140 | +7710 | +10280 | +12850 | +15420 | +15420 |
| Cum. ↑ from GMs | +96 | +192 | +288 | +384 | +480 | +576 | +576 |
| JEI-DNN | 414.3M | 538.2M | 662.1M | 786M | 909.9M | 1034M | 1158M (↑ 0.003% ) |

Table 3: Comparison of the number of Mul-Adds per layer between the base T2T-ViT-14 backbone and JEI-DNN. The middle rows show the net increase due to augmenting with IMs and gates.

| | $IC^1$ | $IC^2$ | $IC^3$ | $IC^4$ | $IC^5$ | $IC^6$ | $IC^7$ |
|---|---|---|---|---|---|---|---|
| T2T ViT 14 | 626.2M | 947.7M | 1269M | 1590M | 1912M | 2233M | 2555M |
| Cum. ↑ from IMs | +38.5K | +77K | +115.5K | +154K | +192.5K | +231K | +269.5K |
| Cum. ↑ from GMs | +906 | +1812 | +2718 | +3624 | +4530 | +5436 | +6342 |
| JEI-DNN | 626.3M | 947.7M | 1269M | 1591M | 1912M | 2233M | 2555M |
| | $IC^8$ | $IC^9$ | $IC^{10}$ | $IC^{11}$ | $IC^{12}$ | $IC^{13}$ | $IC^{14}$ |
| T2T ViT 14 | 2876M | 3198M | 3519M | 3841M | 4162M | 4483M | 4805M |
| Cum. ↑ from IMs | +308K | +346.5K | +385K | +423.5K | +462K | +500.5K | +500.5K |
| Cum. ↑ from GMs | +7248 | +8154 | +9060 | +9.96K | +10.87K | +11.78K | +11.78K |
| JEI-DNN | 2876M | 3198M | 3519M | 3841M | 4162M | 4483M | 4805M (↑ 0.01% ) |

**Expected Calibration Error (ECE):** To estimate the ECE, we approximate the probability that $\hat{y}$ is correct given a predicted $\hat{p}_i^{\max}$. We evaluate the accuracy over a set of samples $ACC(\mathcal{Q}) \triangleq \frac{1}{|\mathcal{Q}|} \sum_{i \in \mathcal{Q}} \mathbb{1}[\hat{y}_i = y_i]$ and compare it to the average predicted probability $\bar{P}(\mathcal{Q}) \triangleq \frac{1}{|\mathcal{Q}|} \sum_{i \in \mathcal{Q}} \hat{p}_i^{\max}$. The predicted probability values are sorted (denoting $\nu(j)$ as the sorting index such that $\hat{p}_{\nu(j)}^{\max} < \hat{p}_{\nu(j+1)}^{\max}$) and split into $B$ bins following (Guo et al., 2017) to approximate the ECE:

$$ECE \approx \sum_{b=1}^{B} \frac{|\mathcal{Q}_b|}{N} \Big| ACC(\mathcal{Q}_b) - \bar{P}(\mathcal{Q}_b) \Big| \quad \text{where } \mathcal{Q}_b = \{\nu(N(b-1)/B), \ldots, \nu(Nb/B)\}. \quad (19)$$

### 9.3 CONFORMAL INTERVALS

In this section, we present other ways to obtain conformal intervals in the EEDN setting and show that they are all approximately equivalent. One simple approach is to consider all samples as a single group, ignoring where they exited. We can then use the entire validation set to select a single threshold. An alternative is to derive a different threshold for each IM. To do this, we partition the validation set into multiple smaller validation sets, one associated with each exit IM: $\mathcal{V} = \mathcal{V}^1, \cup, \ldots, \cup \mathcal{V}^L$. Each $val^l$ contains only the samples that exit at the $l$-th IM. In determining conformal interval threshold for the $l$-th IM, we then use only the samples in $\mathcal{V}^l$ and the predicted probabilities for that specific IM: $\hat{\mathbf{p}}_\theta^l(\mathbf{x}_i)$. For each sample, we calculate a score $s_i = 1 - \hat{\mathbf{p}}_\theta(\mathbf{x}_i)$. Given the set of scores $\{s_i\}$ and a coverage value $1 - \alpha$, the conformal interval threshold value is determined by estimating the $1 - \alpha$ percentile of $\{s_i\}$.

We construct four methods by combining different choices:

- **General** : We ignore the gating and use the entire validation set to obtain a single $\tau^{\mathcal{V},\alpha}$ value. We use this common threshold to generate the conformal sets for every exit IM. The score value for a given sample is derived from the predicted probabilities of its selected exit gate.

- **IMs** : We compute one threshold value per gate, but we use the entire validation set $\mathcal{V}$. The scores used to compute the threshold for the $l$-th IM are derived from $\hat{\mathbf{p}}_\theta^l(\mathbf{x}_i)$. In this computation, we do not take any account of where a sample exits.

- **Strict gating** : A strict version of what we used in our experiments. We compute one threshold value per IM, using only the samples in the validation set associated with that IM. We derive the

score function using $\hat{\mathbf{p}}_\theta^l(\mathbf{x}_i)$. If no samples in the validation set are assigned to a specific IM, i.e; $|\mathcal{V}^l| = 0$, we set $\tau_l^{\mathcal{V}^l,\alpha} = 0$.

- **Gated**: The version used to produce the experimental results in the paper. We use the strict gating approach, but if $|\mathcal{V}^l| < 20$, we set $\tau_l = \tau^{\mathcal{V},\alpha}$, where $\tau^{\mathcal{V},\alpha}$ is derived using the General gating approach above.

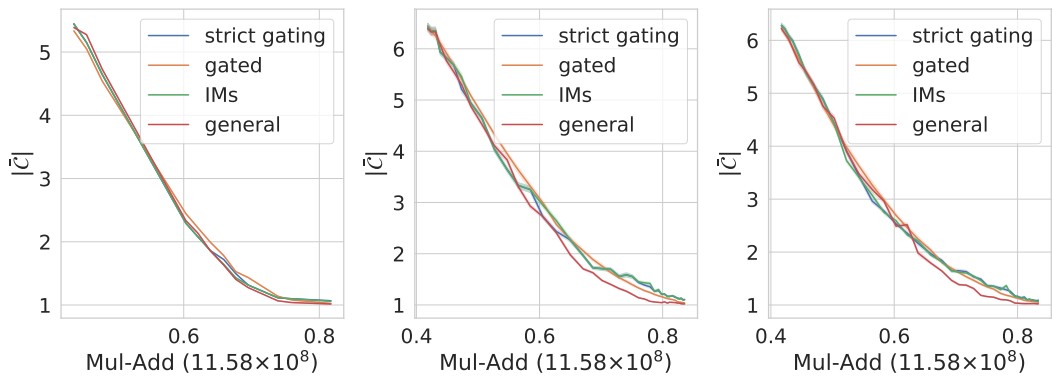

Figure 7: Inefficiency vs Mul-Add on CIFAR10 (t2t-7) of the conformal intervals obtained from the different methods. **(Left)** JEI-DNN, **Middle** Boostnet and **Right** L2W-DEN. The x-axis are scaled by the inference cost of the full model Mul-Add ($IC^L$).

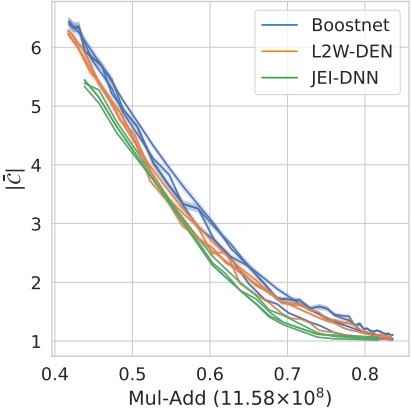

Figure 8: The conformal intervals obtained from all the different methods, for CIFAR10 t2t-vit-7.

Figure 7 shows that all methods are approximately equivalent for each baseline. Figure 8 shows that JEI-DNN achieves the smallest inefficiency for CIFAR10 t2t-vit-7, irrespective of the method chosen to construct confidence intervals. Similar results are obtained for the other datasets. To ensure that the comparison is meaningful, we report the smallest average inefficiency with an empirical coverage that **exceeds** the targeted coverage $1 - \alpha$. To do this, for a targeted $\alpha = 0.05$, we compute the conformal intervals from multiple thresholds, obtained by slowly decreasing the target $\alpha$: $\{\tau^{\alpha=0.05}, \tau^{\alpha=0.045}, \dots\}$ until the empirical coverage is above our target: $1 - \hat{\alpha} > 1 - \alpha$. We then report the inefficiency at that empirical coverage.

## 9.4 SURROGATE LOSS

In this section, we provide the connection between the surrogate problem we introduced for training the $\phi$s and the initial problem as defined in equation 12.

Here, we don't assume that the solutions are unique so we will denote the set of solutions by $\Phi$. The initial problem is then given by:

$$\Phi^* = \arg\min_{\phi} \frac{1}{N} \sum_{i=1}^{N} \sum_{l=1}^{L} C_{\theta^*(\phi)}^{l} P_{\phi}(G = l|\mathbf{x}_i). \tag{20}$$

We replace it by the surrogate problem:

$$\Phi^{sur} = \arg\min_{\phi} \frac{1}{N} \sum_{i=1}^{N} \sum_{l=1}^{L} \mathcal{L}^{CE}(t_i^l, g_{\phi}^l(\mathbf{c}_i^{\leq l})). \tag{21}$$

$$\text{where } t_i^l = \begin{cases} 0 \text{ if } j < l^* \\ 1 \text{ if } j \geq l^* \end{cases} \tag{22}$$

$$l^* = \arg\min_{l} C_{\theta^*(\phi)}^{l}. \tag{23}$$

To motivate this surrogate problem, we show that if there exists a solution $\phi' \in \Phi^*$ s.t. the associated $P_{\phi'}(G = l|\mathbf{x}_i)$s can be equal to each $P_i'$ that minimizes:

$$P_i' = \arg\min_{P} \sum_{l=1}^{L} C_{\theta^*(\phi)}^{l} P_i^l, \left( \text{ with } \sum_{l=1}^{L} P_i^l = 1, P_i^l \geq 0 \right) \quad \forall i \in [N], \tag{24}$$

then we can prove that $\phi'$ will also be an optimal solution for our surrogate problem ($\phi' \in \Phi^{sur}$).

That is, $\phi'$ also minimizes:

$$\frac{1}{N} \sum_{i=1}^{N} \sum_{l=1}^{L} \mathcal{L}^{CE}(t_i^l, g_{\phi}^l(\mathbf{c}_i^{\leq l})). \tag{25}$$

For a given sample $i$, in order to find the probability vector $P_i'$ that minimizes $\sum_{l=1}^{L} C_{\theta^*(\phi)}^{l} P_i^l$, we need to allocate all the probability mass of $P$ to the gate $l^*$ with the lowest cost. That is, the solution to Eqn 24 is given by:

$$P_i'^l = \begin{cases} 1 & \text{if } l = l^* = \arg\min_j C_{\theta^*(\phi)}^{j} \\ 0 & \text{o.w.} \end{cases} \quad \forall i \in [N] \tag{26}$$

From our assumption, we assume that a solution $\phi'$ satisfies $P_{\phi'}(G = l|\mathbf{x}_i) = P_i'^l$. We then show that such a $\phi'$ would then also be a solution to our surrogate problem.

In our surrogate problem, the associated gates $g_{\phi'}^l$ have to satisfy:

$$g_{\phi'}^l = \begin{cases} 1 \text{ if } l = \arg\min_j C_{\theta^*(\phi)}^{j}, \\ 0 \text{ if } l < \arg\min_j C_{\theta^*(\phi)}^{j}. \end{cases} \tag{27}$$

Proof:

- For the trivial case where $l^* = 1$, if $g_{\phi'}^1 = 1$, then $P_{\phi'}(G = 1|\mathbf{x}_i) = 1$ by definition of $g_{\phi'}^1(\mathbf{c}^1)$ and $P_{\phi'}(G = l|\mathbf{x}_i) = min(g_{\phi}^l(\mathbf{c}_i^{\leq l}), 0) = 0$ for $l > 1$.

- Suppose $l^* > 1$, then $g_{\phi'}^l = 0$ for all $l < l^*$. This results in $P_{\phi'}(G = l|\mathbf{x}_i) = min(0, 1 - \sum_{j=1}^{l} 0) = 0$. Once we reach $g_{\phi'}^{l*} = 1$, this results in $P_{\phi'}(G = l^*|\mathbf{x}_i) = min(1, 1 - \sum_{j=1}^{l^*-1} 0) = 1$ and $P_{\phi'}(G = l|\mathbf{x}_i) = min(g_{\phi}^l(\mathbf{c}_i^{\leq l}), 1 - 1) = 0$. This concludes the proof.

Hence, since the optimized gates $g_{\phi'}^l$ are given by equation 27 which corresponds to the solution of our surrogate tasks, we have shown that those $\phi'$ would be shared as the solution of both problems.

## 9.5 ALGORITHM

---

**Algorithm 1** Training

---

1: **Input:** num. of gates $L$, cost importance $\lambda$ parameter, num of epochs $E$, num of warm-up epochs $WE$, bilevel switch count bi_switch
2: STATE=WARMUP
3: **for** $we$ in $\{1, \ldots, WE\}$ **do**
4:    **for** $\mathbf{x}_i, y_i$ in batch **do**
5:       **for** $l$ in $\{1, \ldots, L\}$ **do**
6:          $\mathcal{L}^l_{\theta_l}(x) = CE(p^l_\theta(x), y)$ (optimize each IM)
7:          Update the $\theta_l$ parameters
8:       **end for**
9:    **end for**
10: **end for**
11: STATE=GATE
12: **for** $e$ in $\{WE, \ldots, E\}$ **do**
13:    **for** $\mathbf{x}_i, y_i$ in batch **do**
14:       **if** STATE = CLASSIFIER **then**
15:          Get probability of exiting at each gate l: $P_\phi(G = l | \mathbf{x}_i)$
16:          Optimize the weighted loss $\mathcal{L}^{in}_\theta = \sum_{l=1}^L P_\phi(G = l | \mathbf{x}_i) \mathcal{L}^{CE}(y_i, \hat{\mathbf{p}}^l_{\theta,i})$
17:       **else if** STATE = GATE **then**
18:          Compute the inference cost at each gate l: $IC^l_{\text{norm}}$
19:          Compute the accuracy cost at each gate l: $\mathcal{L}^{CE}(y_i, \hat{\mathbf{p}}^l_{\theta^*,i})$
20:          Store the total cost of that gate: $C^l_{\theta^*(\phi)} = \left( \mathcal{L}^{CE}(y_i, \hat{\mathbf{p}}^l_{\theta^*,i}) + \lambda IC^l_{\text{norm}} \right)$
21:          Compute the best gate $l* = \arg\min_l C^l_{\theta^*(\phi)}$ )
22:          Optimize $\mathcal{L}^{out}_\phi = \sum_{l=1}^L \frac{\partial \mathcal{L}^{CE}(t^l_i, g^l_\phi(\mathbf{c}^{\leq l}_i))}{\partial \phi}$ ( equation 16)
23:       **end if**
24:       **if** batch idx = bi_switch **then**
25:          switch from STATE GATE to CLASSIFIER or from STATE CLASSIFIER to GATE
26:       **end if**
27:    **end for**
28: **end for**

---

## 9.6 ABLATION STUDY

To verify that our joint training procedure is needed, we conduct an ablation study on the CIFAR10 dataset where we adapt our algorithm to the two decoupled settings:

1. **Learnable GMs, fixed IMs:** We use the same warm-up procedure, but avoid bi-level optimization, i.e., after warm-up we only optimize the gates' parameters, $\phi$. We follow Algorithm 1 exactly, but never enter the $CLASSIFIER$ state. We train for the same total number of epochs.

2. **Threshold GMs:** We also adapt our algorithm to the other setting of threshold GMs. We remove the learnable gates from the architecture. We first train the IMs on all the data and then use the threshold gating algorithm used by Yu et al. (2023) and Han et al. (2022b).

The importance of joint training is illustrated Figure 9. The **Learnable GMs, fixed IMs** approach is less stable but slightly outperforms the **Threshold GMs** approach.

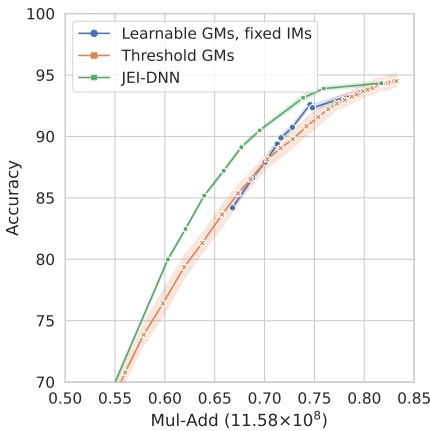

Figure 9: Ablation study on CIFAR10 (t2t-7). Jointly training the GMs and IMs is crucial; both decoupled approaches result in a performance decline.

## 9.7 OPTIMIZATION OF $\mathcal{L}^{in}$

The complete gradient to optimize the inner gradient of our bi-level formulation is given in equation 14:

$$\frac{\partial \mathcal{L}^{in}}{\partial \theta} = \frac{1}{N} \sum_{i=1}^{N} \sum_{l=1}^{L} \frac{\partial \mathcal{L}^{CE}(y_i, \hat{\mathbf{p}}_\theta^l(\mathbf{x}_i))}{\partial \theta} P_\phi(G = l|\mathbf{x}_i) + \frac{\partial G_\phi(m(\theta, \mathbf{x}_i))}{\partial \theta} C_{\theta(\phi)}^l \tag{28}$$

In our methodology, we mention that in practice, we ignore the right term of the gradient, and approximate the gradient using the term that corresponds to the gradient of a weighted cross entropy loss:

$$\frac{\partial \mathcal{L}^{in}}{\partial \theta} = \frac{1}{N} \sum_{i=1}^{N} \sum_{l=1}^{L} \frac{\partial \mathcal{L}^{CE}(y_i, \hat{\mathbf{p}}_\theta^l(\mathbf{x}_i))}{\partial \theta} P_\phi(G = l|\mathbf{x}_i) + \frac{\partial G_\phi(m(\theta, \mathbf{x}_i))}{\partial \theta} C_{\theta(\phi)}^l \tag{29}$$

$$\frac{\partial \mathcal{L}^{in}}{\partial \theta} \approx \frac{1}{N} \sum_{i=1}^{N} \sum_{l=1}^{L} \frac{\partial \mathcal{L}^{CE}(y_i, \hat{\mathbf{p}}_\theta^l(\mathbf{x}_i))}{\partial \theta} P_\phi(G = l|\mathbf{x}_i). \tag{30}$$

We made that decision following our observation that both approaches yield comparable outcomes, but opting for the complete gradient leads to a slower convergence. We denote the JEI-DNN version using the full gradient equation 29 as **JEI-DNN-FULL**. Figure 10 shows the test accuracy of both models during training; we can see that both reach the same test accuracy, but that JEI-DNN reaches it considerably faster.

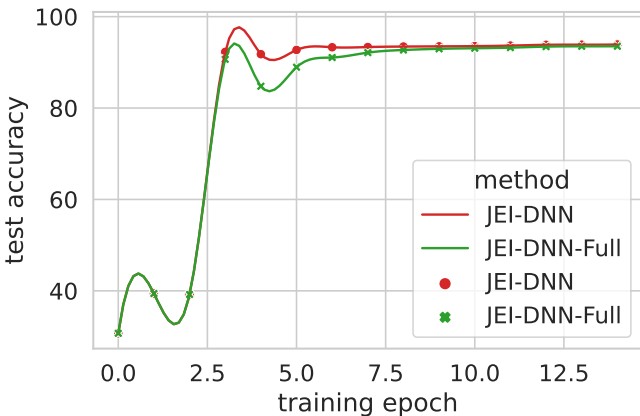

Figure 10: Test accuracy during training of both our proposed JEI-DNN and JEI-DNN-FULL.

## 9.8 ADDITIONAL RESULTS

We include additional performance versus inference cost curves in Figure 11, additional calibration versus inference cost curves in Figure 12 and inefficiency versus inference cost curves in Figure 13. The same trends as presented in the main paper can be observed.

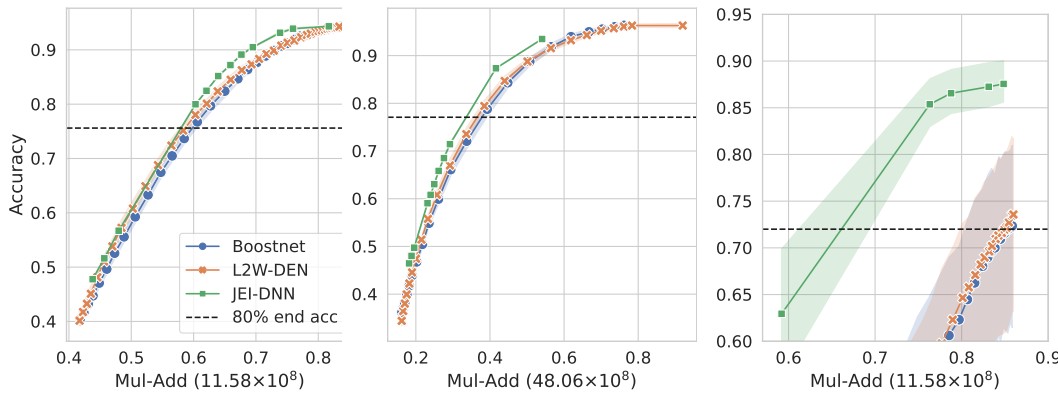

Figure 11: Accuracy vs Mul-Add of **(Left)** CIFAR10 (t2t-7), **Middle** CIFAR10 (t2t-14) and **Right** CIFAR100LT (t2t-7). The x-axes are scaled by the inference cost of the full model Mul-Add ($IC^L$).

## 9.9 UNCERTAINTY STATISTICS

In this section we provide details concerning the uncertainty statistics that are used as input to our gate functions $g_\phi$. We represent the gate input as $m(\theta, \mathbf{x}_i) = [\hat{p}_i^{max}(\mathbf{x}_i), h(\mathbf{x}_i), h_{pow}(\mathbf{x}_i), mar(\mathbf{x}_i)]^T$ where:

- $\hat{p}_i^{max}(\mathbf{x}_i) = \max(\hat{\mathbf{p}}_i)$ is the maximum predicted probability;

- $h(\mathbf{x}_i) = \sum_k \hat{\mathbf{p}}_{\theta,i}^{l,(k)} \log \hat{\mathbf{p}}_{\theta,i}^{l,(k)}$ is the entropy of the predictions;

- $h_{pow}(\mathbf{x}_i) = \sum_k \tilde{\mathbf{p}}_{\theta,i}^{l,(k)} \log \tilde{\mathbf{p}}_{\theta,i}^{l,(k)}$ where $\tilde{\mathbf{p}}_{\theta,i}^{l,(k)} = \frac{(\hat{\mathbf{p}}_{\theta,i}^{l,(k)})^2}{\sum_{k'}(\hat{\mathbf{p}}_{\theta,i}^{l,(k')})^2}$ are scaled predictions (temp. = 0.5);

- $mar(\mathbf{x}_i) = p_i^* - p_i'$ for $k_i^* = \arg\max_k \hat{\mathbf{p}}_{\theta,i}^{l,(k)}$, $p_i^* = \hat{\mathbf{p}}_{\theta,i}^{l,(k_i^*)}$, and $p_i' = \max_{k' \neq k_i^*} \hat{\mathbf{p}}_{\theta,i}^{l,(k')}$. This is the difference between the two most confident predictions.

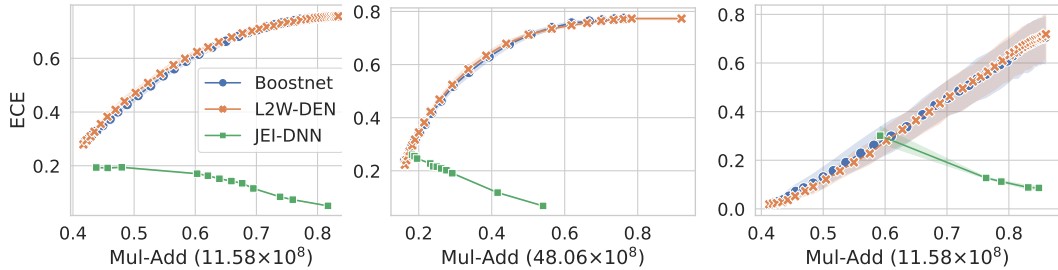

Figure 12: Calibration error vs Mul-Add of **(Left)** CIFAR10 (t2t-7), **Middle** CIFAR10 (t2t-14) and **Right** CIFAR100LT (t2t-7). The x-axes are scaled by the inference cost of the full model Mul-Add ($IC^L$).

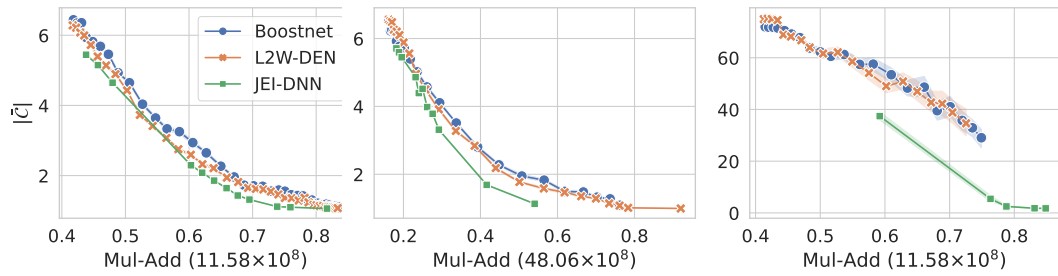

Figure 13: Inefficiency vs Mul-Add of **(Left)** CIFAR10 (t2t-7), **Middle** CIFAR10 (t2t-14) and **Right** CIFAR100LT (t2t-7). The x-axes are scaled by the inference cost of the full model Mul-Add ($IC^L$).

## 9.10 RELATED WORK

In this appendix, we provide a more extensive and detailed discussion of related work, touching on some topics that are more tangentially related.

**Architectures with threshold GMs**  The majority of the literature on early-exit neural networks has primarily focused on designing architectures that are amenable to efficient and effective early-exit. Generally, in these approaches, the decision to exit at an earlier branch is taken by comparing confidence scores extracted from the IMs to predetermined thresholds.

The idea is first introduced by BranchyNet (Teerapittayanon et al., 2016), which augments a network with early-exit branches and trains the augmented model end-to-end, by summing a weighted loss of all IMs. Training the whole network end-to-end can bring complications as earlier IMs negatively impact the performance of later IMs. With this in mind, MSDNet (Huang et al., 2018) uses dense connections to reduce the influence of early IMs on later layers. The performance of earlier IMs is also improved by using representations of varying scales to train with coarser features. In RANNet (Yang et al., 2020), spatial redundancy of the input is exploited, with higher resolutions used only for the more difficult samples. In SDN (Kaya et al., 2019), off-the-shelf networks are augmented with intermediate IMs. Two training schemes are explored: freezing the backbone while training only the early exit branches or training the full network. The experimental results suggest that training the full network works better since the frozen backbone is optimized for the last layer. The interference of the early-exit branches on later layers is addressed via a weighted loss scheme with weights that depend on the position of the exit. More recently, Han et al. (2023b) proposed the Dynamic Perceiver, demonstrating that any CNN-based state-of-the-art vision backbone can be augmented with a parallel computation path that is optimized for early classification. The introduction of such a parallel branch can improve early-exiting while reducing the interference of early IMs on the performance of later IMs.

Table 4: Classification of EEDN problem settings and approaches. A tick mark indicates that a method is capable of providing this feature. **Fixed backbone:** The method can be applied to off-the-shelf backbones without re-training, i.e., it does not require a tailored EEDN backbone. **Learnable GMs**: The gates can be trained. Most methods use threshold GMs. **Train IMs**: The IMs are trained. **Adaptive IM training**: The training is adapted to different IMs, either by introducing weights or training with different subsets of the training data. **Joint Training**: Gate and IM parameters are jointly learned. * Indicates the baselines we include in our work.

| Model | Fixed backbone | Learnable GMs | Train IMs | Adaptive IM Training | Joint Training |
|---|---|---|---|---|---|
| BranchyNet (Teerapittayanon et al., 2016) | | | ✔ | | |
| * MSDNet (Huang et al., 2018) | | | ✔ | | |
| SDN (Kaya et al., 2019) | ✔ | | ✔ | | |
| (Wang et al., 2020) | | | ✔ | | |
| (Wang et al., 2021) | | | ✔ | | |
| (Chen et al., 2023a) | | | ✔ | | |
| DynPerceiver Han et al. (2023b) | | | ✔ | | |
| ReX: (Qian et al., 2022) | | | ✔ | | |
| * IMTA (Li et al., 2019) | | | ✔ | | |
| Meta-GF (Sun et al., 2022) | | | ✔ | | |
| (Bolukbasi et al., 2017) | | ✔ | ✔ | | |
| Predictive Exit (Li et al., 2023) | ✔ | ✔ | | | |
| EPNet Dai et al. (2020) | | ✔ | ✔ | | |
| PTEENet (Lahiany & Aperstein, 2022) | ✔ | ✔ | | | |
| EENet (Ilhan et al., 2023) | ✔ | ✔ | | | |
| Block-Dependent (Han et al., 2023a) | | | ✔ | ✔ | |
| *BoostedNet(Yu et al., 2023) | ✔ | | ✔ | ✔ | |
| * L2W(Han et al., 2022b) | ✔ | | ✔ | ✔ | |
| * DiffBranch (Scardapane et al., 2020) | ✔ | ✔ | ✔ | ✔ | ✔ |
| **JEI-DNN** | ✔ | ✔ | ✔ | ✔ | ✔ |

Another line of work has investigated ways of improving the training and performance of EE architectures by tackling the issue of conflicting gradient updates on the shared backbone parameters. These arise because the training procedure is attempting to simultaneously optimize multiple different exits. In IMTA (Li et al., 2019), the gradients computed when optimizing the intermediate IMs are rescaled based on their position in the network to ensure they have a bounded scale, reducing their variance. Additionally, knowledge distillation is employed to help earlier IMs learn from deeper ones. Meta-GF Sun et al. (2022) also addresses conflicting gradient updates, but rather than using position-based rescaling, the scaling factors are learnt via a meta-learning formulation. The motivation is that in large over-parameterized models, certain parameters of the backbone are more relevant to some exits than others. Therefore, Meta-GF takes into account each parameter's influence on the various exits before scaling the respective gradients.

Other research efforts have focused on customizing EEDNs to a specific type of backbone architecture. Wu et al. (2018) introduced an EEDN architecture customized for Residual networks, while Chen et al. (2018) did the same for CNNs.

Fundamentally, all these studies rely on confidence score thresholds that are set as hyperparameters to decide whether to early-exit. Most recent EEDN advances fall into this category of a EEDN-tailored architecture paired with either threshold GMs (Wang et al., 2020; Han et al., 2023b; Wang et al., 2021; Chen et al., 2023b) or other types of fixed gating mechanisms (Qian et al., 2022).

While most of these studies concentrate on image classification tasks, EEDNs have also seen expansion into other domains, including language models (Zhou et al., 2020; Schuster et al., 2022), as well as video frame classification (Seon et al., 2023). Graves (2016) introduces an RNN architecture where the model is trained to predict and to learn the number of computational steps required, which can be framed as an end-to-end EEDN model.

EEDN research efforts have also explored how to fully leverage the additional computations performed by the **previous** gates and inference modules (Jiang & Mu, 2021; Wolczyk et al., 2021). For example, Wolczyk et al. (2021); Liu et al. (2022) and Passalis et al. (2020) all combine the outputs

of the IMs using ensemble methods to enhance performance. Similarly, Qendro et al. (2021) also employ ensembling of IMs but target the uncertainty characterization of the overall network rather than inference cost. Of particular interest among the ensembling techniques is the geometric ensembling procedure of Wolczyk et al. (2021). When combining the predictions of the IMs, the output probability at IM $l$ is raised to the power $w_l$, where $w_l$ is a learnable parameter. This allows the model to learn to place less emphasis on the predictions from the weaker IMs at lower layers during the fusion process. In our proposal, we do not form ensembles, but we do learn the exit probabilities. If we interpret each IM softmax output as a probability distribution across labels, then the overall label distribution for any given sample is a mixture of the individual IM label distributions, where the mixture weights are the learned exit probabilities. Thus, although our learning procedure does not perform ensembling for any specific classification decision, its probabilistic output can be viewed as being drawn from an ensemble of classifiers.

**Learnable GM; pre-trained and frozen backbone and IMs**  Differing from the dedicated architecture and **threshold GMs** methods described above, another class of EEDN treats the IMs and backbone architecture as a fixed, frozen component, and focuses on training the GMs. In some cases, the pretraining of the IMs is included in the methodology. In (Bolukbasi et al., 2017), the learning is performed in two separate phases. First, the IMs are trained with the full training set and then the IMs are frozen and the gates are trained. The gates are trained sequentially starting from the deepest layer to the shallowest, allowing the gate training objective to be defined recursively. In EP-Net (Dai et al., 2020) the task of training the gates after freezing the IMs is formulated as a Markov decision process. The methodology involves designing a substantial, trainable neural network of multiple layers for the gate mechanism. This was critized by following works as being wasteful, imposing an unacceptable inference overhead. PTEENet(Lahiany & Aperstein, 2022) parameterizes the gates and directly learns the exit decisions using sigmoid neural networks. EENet (Ilhan et al., 2023) adopts a different strategy by formulating the early-exit policy as a multi-objective optimization problem. Karpikova et al. (2023) design learnable gates, and specifically focus on confidence metrics tailored to images. Finally, there are tangential works such as (Li et al., 2023) that consider the related problem of optimizing gate placement for energy efficiency.

All of these models learn gating functions after having trained and frozen the IMs at every branch. While this two-phase procedure simplifies training, it ignores the coupling between gates and IMs — the gating decisions change the distributions of samples presented to the IMs, implying that parameters should be re-learned.

**Addressing the train-test mismatch**  Two recent works have proposed architecture-agnostic training procedures to address the aforementioned train-test mismatch. In BoostNet (Yu et al., 2023), the training procedure is inspired by gradient boosting. Prediction reweighting is employed so that deeper IMs are trained with prediction reweighting to induce an emphasis on samples that are incorrectly classified by the earlier IMs. Han et al. (2022b) cast the problem as a meta-learning task where a *weight prediction network (WPN)* is trained jointly with the early-exit network to predict the most likely exit of a sample. The predictions of the WPN are used to assign weights to the losses of the different IMs. Consequently, easy samples should have a reduced influence on deeper IMs while harder samples contribute more to their training. This architecture is related to ours but the training procedures differ in multiple important ways. Most importantly, we do not rely directly on thresholding uncertainty metrics for early exiting. We show that the uncertainty metrics are not a good proxy for accuracy for earlier IMs due to miscalibration. In practice, the inclusion of the weight prediction network does not close the train-test gap because it is trained using poorly calibrated labels. By jointly training the IMs and the GMs via bi-level optimization, our approach provides more reliable uncertainty characterization.

Scardapane et al. (2020) propose differentiable branching, an approach where gates and IMs are jointly learned. The differentiable branching uses a straightforward modeling approach for the gate exit probabilities and optimizes all parameters jointly; as a result, training is highly unstable.

Chen et al. (2020) propose a fully trainable architecture featuring a learnable policy component that is responsible for determining the number of layers used for inference. Although Chen et al. (2020) do not target inference efficiency, their framework shares similarities with ours. The gating mechanism is modelled as a product of independent exiting events. However, the inference cost is not incorporated in the training objective. The gate exit distribution is controlled by an entropy

regularization regularization; as a result, the procedure in (Chen et al., 2020) cannot be used to construct an effective EEDN.

This concludes our review of the most relevant EEDN contributions. We include a categorization of the works in Table 4. As is evident from the table, our approach is the only one that (i) can be applied to any layer-based off-the-shelf backbone network, without re-training the backbone; (ii) has learnable GMs and IMs; and (iii) jointly trains both the GMs and the IMs.

**Sparsely-gated mixture-of-experts (MoE)**   A closely related field to our problem setting is the Mixture of Experts (MoE) framework (Shazeer et al., 2017; Fedus et al., 2022; Du et al., 2022). In this framework, the network also employs a gating mechanism to make adaptive decisions based on the input it receives. However, in the MoE setting, the gating decisions are made across layers to incorporates multiple experts or weights (Shazeer et al., 2017; Du et al., 2022; Fedus et al., 2022). The goal is not to tailor inference time based on the input complexity but rather to enable the use of increasingly larger models by maintaining constant inference costs with sparse activation strategies. In this field, it is also considered an important problem to bridge the gap between the gating decisions and the inference modules (Gupta et al., 2022; Zhu et al., 2022). The interpretability potential of such architectures is also highlighted in (Pavlitska et al., 2022).

**Online learning**   In online learning with deep neural networks the aim is to jointly learn representations and a suitable architecture. However, given that the data is streamed serially during training, the optimal depth of the network may change as more data is accumulated. To address this, Sahoo et al. (2018) proposed to incorporate learnable GMs that produce a convex combination of IMs to construct a prediction that is a weighted average of all IM outputs and the final layer output. The GM training employs the hedge backpropagation algorithm proposed by Freund & Schapire (1997).

### 9.11 JEI-DNN ON OTHER BACKBONE ARCHITECTURES

As previously mentioned, the JEI-DNN training procedure can be applied to many different types of architecture. To demonstrate this, we use our procedure on the GRIT (Ma et al., 2023) backbone. GRIT is a transformer-based model for graph data that achieves state-of-the-art performance on the most common graph benchmark datasets proposed by Dwivedi et al. (2022). As before, we augment the backbone with IMs and lightweight gates. Figure 14 shows that out procedure leads to a better accuracy-cost combination than Boostnet when trained on Super-Resolution-MNIST graph dataset. This demonstrates the flexibility of our training procedure, showing that it can achieve efficient inference on a variety of deep architectures addressing different learning domains.

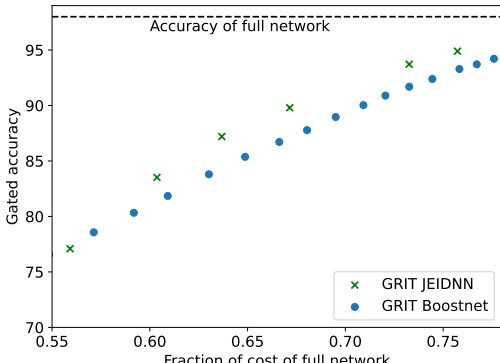

Figure 14: Comparison of the JEI-DNN training scheme and Boostnet on the GRIT backbone on the super-resolution graph MNIST dataset.

### 9.12 ADDITIONAL DATASET: IMAGENET

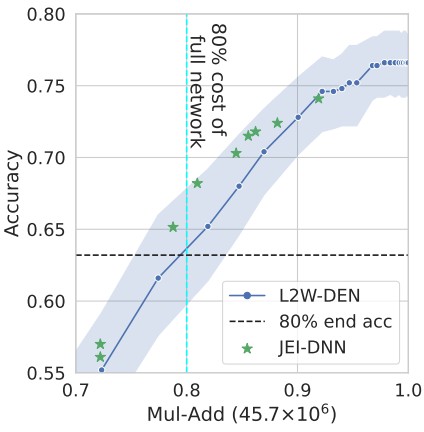

Figure 15: Accuracy vs Mul-Add of ImageNet (t2t-14).

We provide the additional performance versus inference cost curves for the ImageNet dataset (Deng et al., 2009)(see Appendix 9.1.1 for details) using the vision transformer T2T-ViT-14. Since this model was trained on ImageNet, no transfer learning is needed. We follow the general procedure as described in our paper; we augment the backbone with gates and intermediate IMs at every layer. We generate results by varying the inference cost parameter $\lambda$ over the range 0.01 to 10. Rather than sample $E^l \sim p(E^l|\mathbf{x})$ to decide whether to exit, we use the deterministic decision $P(E^l) > 0.5$.

In Figure 15, we compare our method with the most competitive baseline (L2W-DEN) for T2T-ViT-14. In the main paper, we also provide results for ImageNet but using the specialized early-exit architecture Dynamic Perceiver as a backbone in Figure 6 **(right)**.

We observe that by using the JEI-DNN training strategy we achieve an improved trade-off between accuracy and inference cost for T2T-ViT-14 compared to L2W-DEN, the best-performing architecture-agnostic baseline training strategy. For the Dynamic Perceiver backbone, the incorporation of JEI-DNN training lead to a reduction of inference cost of approximately 1-2 percent for operating points in the range from 50-65% of the inference cost of the original network. In this region, the preserved accuracy ranges from 85-95% of the accuracy of the full network. The Dynamic Perceiver backbone has only four IMs, with three located relatively close to the final layer in the architecture. This imposes some limits on the flexibility of the exit decisions and means that it is more challenging for the JEI-DNN training to achieve an improvement for operating points where the accuracy is close to the original accuracy.

The decrease in accuracy of JEI-DNN is steeper when the backbone is a general architecture such as T2T-ViT-14 as opposed to an architecture tailored for early-exit (Dynamic Perceiver). JEI-DNN with T2T-VIT-14 sees its performance decrease to around 75% of its full performance with 80% of its total inference cost, whereas JEI-DNN with the DynPerceiver can maintain 92% of its full performance with only 60% of its total cost.

## 9.13 ADDITIONAL BASELINE

We provide preliminary results for an additional baseline, DiffBranch (Scardapane et al., 2020), in Figure 16, which we significantly outperform. The performance of (Scardapane et al., 2020) is abnormally low, and those results persisted even after our best effort at hyperparameter tuning and different possible implementations. Unfortunately, as the code is not provided and implementation details were missing from the paper, we are unable to confirm that our implementation and therefore our results are correct. For these results, to ensure a fair comparison, we employ the same warm-up procedure as used by our method. Without it, the IMs of DiffBranch (Scardapane et al., 2020) were not able to reach a reasonable accuracy. We also used the same parameterization of the IMs and GMs as JEI-DNN to adapt the architecture to the t2t-vit-7.

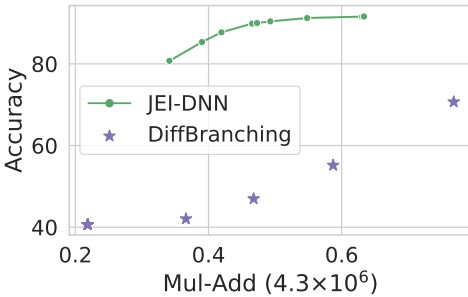

Figure 16: Accuracy vs Mul-Add of SVNH (t2t-7) of the DiffBranch baseline. The x-axes are scaled by the inference cost of the full model Mul-Add ($IC^L$).

## 9.14 ADDITIONAL ACCURACY DECOMPOSITION ANALYSIS

In this section, we offer additional insights into how our algorithm behaves across a wide range of inference costs, illustrated through supplementary decomposition figures. In our main paper, Figure 3 corresponds to $\lambda = 1$ for the CIFAR100 experiment with t2t-vit-14 (Figure 2). Here, we provide in Figure 17 the same decomposition for $\lambda = 3$ , which imposes a greater penalty on computation and leads to earlier exiting behaviour. Figure 18 shows the decomposition for $\lambda = 0.5$, which imposes a small penalty for additional computation, and leads to more samples exiting at later layers.

We can see that for all $\lambda$ values, we obtain a similar trend that shifts from early gates to later gates based on the inference cost. For the larger value of $\lambda$ in Figure 17, JEI-DNN starts exiting samples at gate 3 and heavily uses the gates 4, 5, 6 and 7. For a smaller penalty on computation in Figure 18, JEI-DNN starts to exit samples much later, at gate 5, and mainly uses the gates 7, 8, 9, 10 and 11. These results highlight how the parameter $\lambda$ can be used to control the trade-off between computation during inference and accuracy.

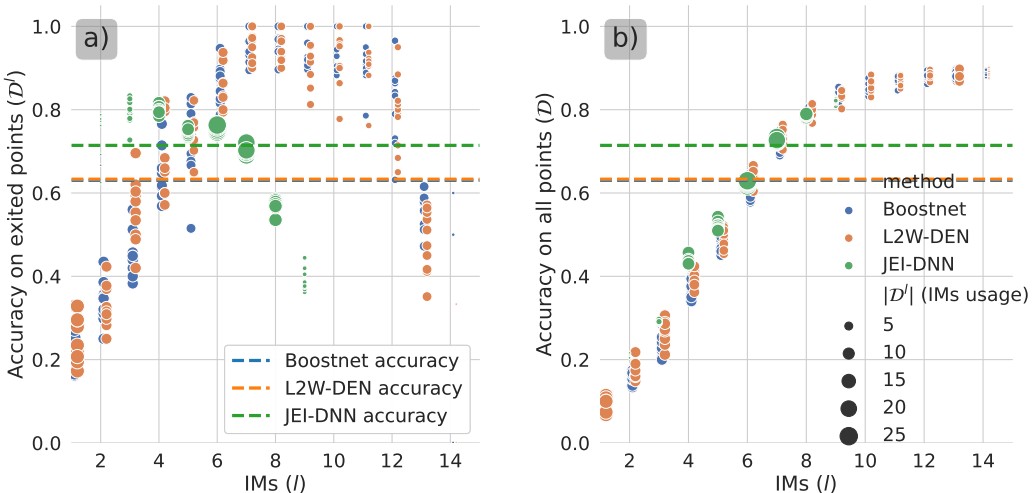

Figure 17: Decomposition of the contributions of the IMs to the final accuracies (depicted by dotted lines) for CIFAR100, with the inference cost $\lambda = 3$. **a)** Accuracy of each IM, $f_\theta^l$, evaluated only on their exited samples ($\mathcal{D}^l$). **b)** Accuracy of each IM, $f_\theta^l$, on the full dataset $\mathcal{D}$. The size of a circle is proportional to the number of samples exited for a trial.

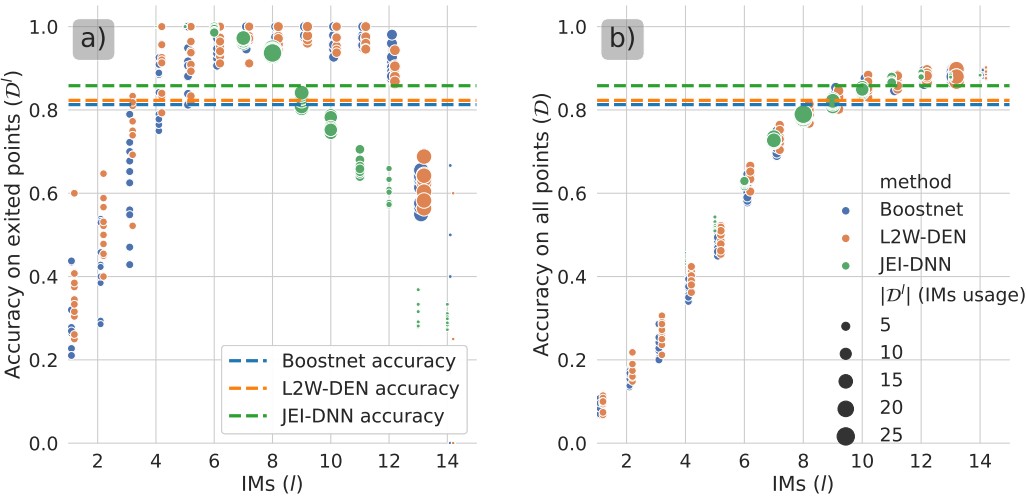

Figure 18: Decomposition of the contributions of the IMs to the final accuracies (depicted by dotted lines) for CIFAR100, with the inference cost $\lambda = 0.5$.

### 9.15 OPTIMIZING THE GATES: BINARY CLASSIFICATION SURROGATE TASKS

In the section **Optimizing the gates** of our paper, we describe a surrogate learning task that we use to train our gating mechanism. In short, we train each of our $g_\phi^l$ modules as a binary classifier, and we set the target $t^l$ based on the the relative cost of each gate $C_{\theta*}^l$. We first find the index of the optimal gate $l^* = \arg\min_l C_{\theta*}^l$ and set its target to 1 $t^{l^*} = 1$, then we set the targets of all the previous gates to 0, and the targets of all the subsequent gates to 1.

The intuition behind this is that, if a sample misses the exit at the optimal IM $l^*$, the next best exit is the following one, at $l^* + 1$. Underpinning this is an assumption that the accuracy of the IMs tends to be monotonically increasing for all samples as the layer increases. The assumption would be wrong if, the best exit is at $l^*$, but then the performance of the IM at the next gate is very poor and the IM should be avoided. (The desired target pattern over 6 gates might then look like $t = 0, 0, 1, 0, 1, 1$, for example).

An alternative strategy is to only set a gate's target to $1$ if there is no later IM that reaches a lower cost (considering both accuracy and inference overhead). So instead of our strategy, that we call ExitSubsequent:

$$t^l = \begin{cases} 1 & , \quad l = l^* \\ 1 & , \quad l > l^* \\ 0 & , \quad l < l^* \end{cases} \tag{31}$$

we could use this other strategy, that we call ExitIfMin:

$$t^l = \begin{cases} 1 & , \quad l = l^* \\ 1 & , \quad l > l^*, C_{\theta^*}^l < C_{\theta^*}^j \, \forall j > l \\ 0 & , \quad \text{o.w.} \end{cases} \tag{32}$$

We verify that the ExitIfMin strategy yields similar results to our ExitSubsequent strategy by running an experiment on the SVHN dataset with t2t-vit-7. The results can be viewed in Figure 19, where we can see that both approaches give the same results. This confirms that our assumption holds.

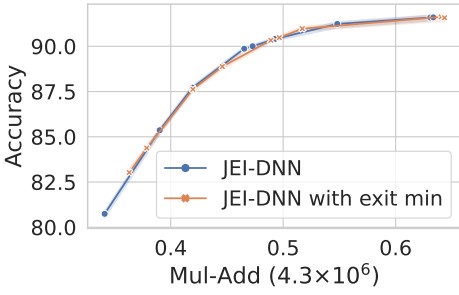

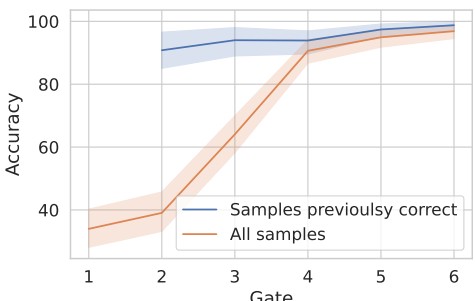

Figure 19: JEI-DNN vs JEI-DNN with ExitIfMin for SVHN dataset with t2t-vit-7 backbone. The x-axis is scaled by the by the full model inference cost, Mul-Add($IC^L$).

Figure 20: Accuracy at each IM for all the samples vs the accuracy at each IM for the samples that were correctly classified by the previous IM. SVHN dataset with t2t-vit-7.

We can also examine the progression of the accuracy at each IM for the samples that were correctly classified by a previous IM. If we see a trend such that once a sample is correctly classified by an IM $l$, it is also correctly classified by all subsequent IMs, then it suggests that it is indeed best to exit the sample as soon as possible. In Figure 20 we compare the conditional accuracy (conditioned on correct classification at the previous IM) with the overall accuracy. We see that the conditional accuracy is higher, and above 90% for all IMs, indicating that our strategy is sensible.

We do note, however, that these results are for an architecture where the intermediate layers are not trained with the goal of providing good features for classification. In a shallow-deep network, the intermediate layers are trained to do so. Kaya et al. (2019) observe that there can be a phenomenon of "overthinking" for such networks, where an early IM can make a correct classification, but later IMs make errors for the same sample. For such a backbone, the ExitIfMin strategy may be preferable.

### 9.16 END-TO-END TRAINING OF JEI-DNN WITH A DEDICATED ARCHITECTURE

Our method is designed to be used on a fixed backbone architecture. This includes enhancing architectures that are tailored for early-exit, as we have shown in our main paper in **Observation** 5.

We can go one step further and consider a fully end-to-end trainable model that trains the backbone, the IMs, and the GMs. Instead of keeping the backbone fixed during training, we can alternate between our JEI-DNN procedure and training the backbone. To present results, we again select as the backbone architecture the Dynamic Perceiver Han et al. (2023b) with MobileNetV3.

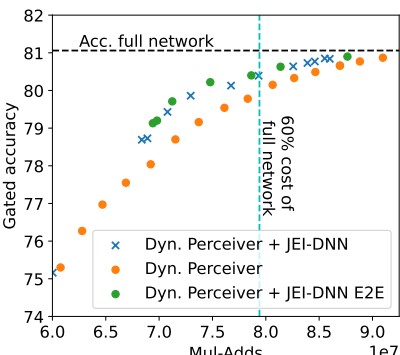

Figure 21: Comparison of Dynamic Perceiver, Dynamic Perceiver trained with the JEI-DNN procedure (Dyn. Perceiver + JEI-DNN) and Dynamic Perceiver trained end-to-end with the JEI-DNN procedure (Dyn. Perceiver + JEI-DNN E2E) on CIFAR100.

The end-to-end combination (Dynamic Perceiver+JEI-DNN) includes learnable gates, learnable IMs, and a learnable backbone. We compare it to the results we have presented with the Perceiver, Perceiver+JEI-DNN, which are both trained for 310 epochs. We train our Perc+JEI-DNN end-to-end by alternating between training JEI-DNN for 2 epochs, then the Perceiver for 1 epoch. Figure 21 shows that this produces similar results to Perceiver+JEI-DNN, but there is a small improvement. For example, if we require 80 percent accuracy (a reduction of only 1 percent from the accuracy of the full network), the end-to-end training means that we can reduce the required mul-adds by a further 2.6%. These promising results indicate that further research into a fully end-to-end training architecture has the potential to be a fruitful research direction.

