# OpenReview forum: "Jointly-Learned Exit and Inference for a Dynamic Neural Network"
_ICLR.cc/2024/Conference — ICLR 2024 poster_

### Official Review · Reviewer_smKZ · 2023-10-26

**Soundness:** 3 good
**Presentation:** 3 good
**Contribution:** 3 good
**Rating:** 8
**Confidence:** 4

**Summary:**

The paper describes a novel mechanism to add trainable early exits (EEs) to a pre-trained neural network. It addresses a common concern of the majority of previous works, where the train procedure and the test procedure of the EEs were mismatched (e.g., jointly training all the early exits during training while thresholding their outputs during test).

In the proposed framework, each EE is associated to a probabilistic gate, and everything is trained jointly with a bilevel optimization problem, where the outer problem is defined over the gates and the inner problem over the EEs. The system is trained to also minimize execution time by training the gates to select the first exit having good accuracy for each pattern.

They also propose a novel conformal prediction (CP) strategy for EE networks, where different thresholds are selected for each exit (since each exit will observe a different subset of values at inference times).

**Strengths:**

The topic of the paper is important, since the method described can be used to reduce inference time (hence, power consumption) of any pre-trained model. The paper is generally well written and easy to read, although some additional visualization could be useful (see below). Results are relatively comprehensive, although I am concerned by the transfer learning from a large dataset (ImageNet) to a smaller, fundamentally similar dataset (CIFAR, also see below).

**Weaknesses:**

I find it hard to fully understand the method before reading through the entire paper. In particular: (a) the abstract has no mention of the paper's contributions; (b) the "Contributions" also does not explain how the method works; (c) there is no visual schema of the model; (d) some important details are described very late, for example "Gate design" is part of the experimental section but it is a crucial design decision. Overall it is understandable, but I think some minor reorganization and some additional visual descriptions (maybe a shortened pseudo-code?) can significantly help the reader.

On the novelty, I think a few methods could be added and discussed more. For example, the zero-time waste model (cited in the paper) has a geometric ensemble of the early exits which is very similar to their gating probability, if I understand correctly. As another example, differentiable branching (https://ieeexplore.ieee.org/document/9054209) considers a trainable exit strategy that also combines the EEs and that can be trained end-to-end with the rest of the network. Some of these methods can be also added to the experimental comparison.

Concerning the last point, I am curious about transfer learning from a large, generalist dataset (ImageNet) to a small dataset which is basically in the same domain (CIFAR-10). Would it not make more sense to test directly on ImageNet? Improvements would be directly comparable with the original pre-trained model.

EDIT AFTER REBUTTAL: the authors have added several baselines and improved the presentation of the method. I have raised my score to 8.

**Questions:**

Apart from the topics discussed above, I have a general question about the bilevel task. Can you clarify why this is needed? There are techniques to train probabilistic blocks (e.g., reparameterization, score function estimators), which would allow to train all blocks simultaneously, or I am missing something? Also, what is the overall training time of your model compared to the alternatives?

---

> ### Author Response · Authors · 2023-11-19
> **Response 1/3**
>
> Thank you for the detailed and perceptive review, which has brought a number of issues to our attention, leading to modifications that we believe have improved the paper.
>
> ### 4.1. Reorganization and visual descriptions:
> Thank you for the suggestions to improve the clarity of the paper. Unfortunately, it is forbidden to modify the abstract. We agree that an additional sentence would help to clarify the contribution. However, following your suggestion, we have added a sentence to the introduction in the ''Contributions'' section that provides a high-level description of how the proposed method works. Also in line with your suggestion, we have added Figure 1, which provides a visual schema of the model and describes the core steps of the methodology. In the original version of the paper, we elected to include ''Gate design'' in the experimental section to separate the core components of the methodology from what we considered to be implementation decisions. For example, it is possible to use the softmax output directly rather than construct uncertainty features (this achieves similar performance but at the expense of a larger gate). We have now moved the gate and IM design into the methodology, as recommended.

---

> > ### Author Response · Authors · 2023-11-19
> > **Response 2/3**
> >
> > ### 4.2. Related work: zero-time waste model and differentiable branching:
> > Thank you for this suggestion. We have updated the discussion of the related work in the main paper to refer more prominently to the two indicated works. We have also added a slightly lengthier discussion in the appendix.
> >
> > The ZTW paper by Wolczyk et al. (2021) focuses on learning weights to ensemble the earlier IMs (or ICs in their work). The weights are not actually used for early exiting. To early exit, ZTW uses the same fixed threshold mechanism as the baseline in our paper, as can be seen in their Equation 3. There is a fixed threshold, and the sample exits if the maximum probability of the prediction exceeds it. The setting that ZTW considers is to reuse the IM values in the best way possible. Our setting is to improve the interaction between the gating mechanism and the IM training. As such, we do not view it as a competing baseline, but addressing another problem of early exiting. It would be an interesting extension to incorporate ZTW into our joint learning framework to determine if reusing earlier IM results can improve performance further.
> >
> > There is some similarity between the geometric ensembling employed by Wolczyk et al. and our proposed method, but there is a key difference in that ZTW does not model exit probabilities but instead ensemble weights. When combining the predictions of the IMs, the output probability at IM $l$ is raised to the power $w_l$, where $w_l$ is a learnable parameter. This allows the model to learn to place less emphasis on the predictions from the weaker IMs at lower layers during the fusion process.
> >
> > In our proposal, we do not form ensembles, but we do learn the exit probabilities. If we interpret each IM softmax output as a probability distribution across labels, then the overall label distribution for any given sample is a mixture of the individual IM label distributions, where the mixture weights are the learned exit probabilities. Thus, although our learning procedure does not perform ensembling for any specific classification decision, its probabilistic output can be viewed as being drawn from an ensemble of classifiers.
> >
> > As for the differentiable branching paper (Scardapane et al., 2020), thank you for bringing it to our attention. It is true that the method is highly related, and we should have included it as a baseline. The critical differences with our method are (i) the way in which the exit probability is modelled; and (ii) the use of bi-level optimization. Scardapane et al. model the exit probability in the product form. We found that this made optimization very challenging and the change to a constrained sum (cumulative exit probabilities summing to 1) was critical. The bi-level optimization, although perhaps not essential, leads to a more robust learning behaviour.
> >
> > We have implemented a version of the differentiable branching approach and
> > we include preliminary results in Appendix 14. The results will be finalized in the next few days. Since the code is not available and some key implementation details are missing (it is a 4-page paper), we are not absolutely sure that our implementation matches what Scardapane et al. did in practice. To give the approach the best chance, we incorporate the same warm-up that we use in our procedure. However, we can see that the performance is still not good, and training is very unstable. Indeed, the results in the paper are limited, but they also suggest that the behaviour may be unstable. For example, we see that for VGG-13, 68.8\% of samples exit at IM 6 and 20.3\% at IM 8, but close to zero at IM 7. This bi-modality is unexpected (considering that the IMs are close so the inference cost does not differ by a substantial amount), and if the learning were stable, we would anticipate a smoother exit behaviour as exhibited by our approach and all of the other baselines we have experimented with.

---

> > > ### Author Response · Authors · 2023-11-19
> > > **Response 3/3**
> > >
> > > ### 4.3. Transfer learning concerns and ImageNet:
> > > Transfer learning is indeed an interesting issue to ponder. Our perspective is that the benefits derived from transfer learning should be reflected in the performance of the backbone architecture. It seems less likely that a transfer learning effect would bias how well a dynamic exit learning procedure performs. Both the original backbone and the augmented backbone with IMs and gates benefit from the same effect, so a comparison between their accuracies and inference costs should be valid. Similarly, it is difficult to see why one dynamic exit training procedure would benefit more than another. On the other hand, we can see that it might raise concerns that the performance of the dynamic exit procedure does not generalize to settings where there is not a strong transfer learning effect.
> > >
> > > We completely agree that providing results for Imagenet would eliminate concerns that transfer learning might be introducing bias.  We have commenced experiments on ImageNet and obtained preliminary results. The experiments are ongoing, and we will finalize and report the results at least one day before the end of the response period. Although our introduced gates and IMs are lightweight and have few parameters, training with ImageNet is still time-consuming, even with relatively powerful servers.
> > >
> > > ### 4.4. Need for bi-level optimization:
> > >
> > > All blocks could be trained simultaneously, but we found that training via the bi-level optimization was more stable and yielded better results. We suspect that one characteristic of the problem that is challenging is the strong interconnections and dependencies between the different components. The model has to simultaneously learn the IM parameters and assign probabilities to sample exit points. Changes in exit probabilities can significantly impact the IM model parameters, but during the early stages of training, the determination of exit probabilities is based on inaccurate IMs. We suspect that the space of the loss has many irregularities since small changes can have a significant effect on the optimal solution. Although the judicious use of probabilistic block training strategies can assist with this, experimentally, we found that training the blocks simultaneously was generally unstable. In summary, we do not believe that bi-level optimization is essential, but it does lead to a reasonably stable and robust solution without too many hyperparameters to configure. We tried multiple techniques for the joint learning, but if you have had good experience with a particular method, we would be pleased to learn about it.
> > >
> > >
> > > ### 4.5. Training time:
> > >  In general, all the methods share similar training time, with BoostedNet begin the quickest, then L2W, the JEI-DNN. The training time, as well as the number of parameters, is provided in Table 1 of the Appendix.

---

> > > > ### Comment · Reviewer_smKZ · 2023-11-20
> > > >
> > > > Thanks for the significant work in the rebuttal. I have increased the presentation score to 3 and the overall score to 8 (accept). I have no specific questions left.

---

### Official Review · Reviewer_1vfw · 2023-10-26

**Soundness:** 3 good
**Presentation:** 3 good
**Contribution:** 3 good
**Rating:** 8
**Confidence:** 5

**Summary:**

This paper proposes to jointly learn the backbone parameters and the exiting strategies in a dynamic multi-exit model. Intermediate classifiers are constructed in a pre-trained backbone, and exiting gates are built as a binary classification head. The joint learning problem is formulated as a bi-level optimization problem. Experiments on several small datasets demonstrate that 1) a better trade-off between accuracy and efficiency is achieved; and 2) a better estimation of uncertainty (calibration of classification confidence) is achieved.

**Strengths:**

1. The studied problem is of great interest;

2. The motivation is clear;

3. The proposed method is technically sound;

4. The literature review is comprehensive;

5. The experiments show the effectiveness in both accuracy-efficiency tradeoff and calibrated confidence.

**Weaknesses:**

1. **Lack of experiment on more advanced architectures**. Why did the authors select a T2T-ViT to construct the early-exiting model? Straightforwardly, the joint learning procedure can be directly applied in mature multi-exit models, such as the compared MSDNet, RANet, and the cited Dynamic Perceiver.  To my understanding, Dynamic Perceiver is the most recent work in this field, achieving SOTA performance in dynamic early exiting.

2. **Overclaiming contributions**. Based on the above point, the contribution of this paper should not include a new architecture. It is recommended to summarize the contribution as the joint learning procedure only. I believe this would already be significant enough if the learning method is shown effective on more SOTA architectures and on the ImageNet dataset.

3. **Lack of experiments on ImageNet**. To my understanding, the learning approach does not need to be applied in a downstream task on the toy small-scale datasets. Experiments on ImageNet would be more convincing.

4. **Presentation**. It is recommended to use figures to clearly show the motivation and the method pipeline.

**Questions:**

See weaknesses.

---

> ### Author Response · Authors · 2023-11-19
> **Response**
>
> We would like to express our appreciation for the thorough review, and in particular, the suggestions to conduct experiments with Dynamic Perceiver. We believe that the experimental results have strengthened the paper.
>
> ### 3.1. Lack of experiments on more advanced architectures; Dynamic Perceiver:
> Thank you for the suggestion to experiment on more advanced architectures and, in particular, those tailored for early-exit. It prompted some interesting and fruitful experiments with Dynamic Perceiver. We initially selected T2T-ViT as an example transformer model that performs well on the tasks under study. But we agree that combination with a state-of-the-art early-exit model would be more compelling.
>
> In Observation 5 in the revised paper, we include results where we combine the Dynamic Perceiver with JEI-DNN (our proposed method). We use the Dynamic Perceiver framework to jointly train the backbone   and the IMs and then use JEI-DNN to jointly train the gates and the IMS.  Figure 6 shows that using JEI-DNN achieves an accuracy improvement of approximately 0.5-1\% compared to Dynamic Perceiver on the CIFAR100 dataset when using MobileNetV3 as a backbone, when using 50-60 percent of the mul-add operations that are used in the original architecture. We are conducting similar experiments on ImageNet and hope to report the results before the end of the response period.
>
> ### 3.2. Overclaiming contributions:
> Thank you for pointing this out. Indeed, we agree that we do not propose a new architecture, and we have removed the word ''architecture'' from the claimed contributions, both in the introduction and the conclusion, in the revised paper.
>
> ### 3.3. Lack of experiments on ImageNet:
> We have commenced experiments on ImageNet and obtained preliminary results. The experiments are ongoing, and we will finalize and report the results at least one day before the end of the response period. Although our introduced gates and IMs are lightweight and have few parameters, training with ImageNet is time-consuming, even with relatively powerful servers.
>
> ### 3.4. Presentation:
> Following your recommendation, we have added Figure 1 at the start of the methodology section to help the reader understand the approach.

---

> > ### Comment · Reviewer_1vfw · 2023-11-23
> >
> > After reading the authors' responses, I have increased my rating.

---

### Official Review · Reviewer_CAtU · 2023-10-31

**Soundness:** 3 good
**Presentation:** 3 good
**Contribution:** 3 good
**Rating:** 8
**Confidence:** 4

**Summary:**

This submission introduces JEI-DNN, an early-exit model approach that can be applied on top of off-the-shelf backbones for image classification. The proposed methodology appends light-weight classifiers and trainable gates along the depth of a frozen backbone model and jointly optimises them with a custom and insightfully designed loss function. As a result, a better speed-accuracy trade-off is provided compared to several baselines considering both learnable and non-parametric exit policies, while offering improved prediction uncertainty.

**Strengths:**

-The submission studies a very interesting problem, focusing on the emerging inference setting of input-dependent computation via early-exiting.

-The manuscript offers a beautiful formulation of the examined task and provided solution, that can benefit the community as a principled definition of early-exit models. Additionally, the discussion of the manuscripts findings presents useful insights that can guide practitioners and inspire future research in the direction of EE models.

-Most design choices in the adopted solution are well motivated and backed by practical insights.

-The presented results indicate that the proposed method is effective and achieves a superior speed-accuracy trade-off to a wide set of baselines approaches.

**Weaknesses:**

-A few technical aspects of the paper remain unclear. Specifically, the use of the min operator between two terms in Eq.8 is not fully justified in the manuscript, nor experimentally verified by a relevant ablation.

-The proposed methodology is only evaluated on frozen-backbone CNNs. Although this a practical sub-category of EE-models, further evaluation on end-to-end EE models (jointly training the backbone and exits), which are shown to achieve superior speed-accuracy trade-off would increase the contribution of the paper.

-Although the proposed approach is compared with a wide set of baseline, ImageNet experiments are omitted, rendering the presented results less convincing.

Should the above limitations be satisfactorily addressed, I am inclined to increase my score.

**Questions:**

1. What is the role of the second (1-sum) term in Eq.8 ? Potentially, its use could be validated through an ablation experiment.

2. How would the method perform on end-to-end trainable early-exit models? Would the training of the gates affect the convergence of the overall model? Should the method be considerably adjusted to be applicable in this setting?

3. Is the proposed approach equally effective on the most commonly used ImageNet-1K benchmark?

Minor Comments:
Sec3: The term "IC-Only training" is widely used in the community (including the cited survey paper) to denote "Intermediate-classifier only training", rather "than inference cost-only training" as stated in the manuscript

Post-Rebuttal Edit: Following the clarification on Q1 and additional results included in the updated version of the manuscript for Q2 and Q3, I am increasing my score from 6(=BA) to 8(=A).

---

> ### Author Response · Authors · 2023-11-19
> **Response**
>
> Thank you for your careful and insightful review of the paper and for your positive comments concerning its strengths. Your constructive criticisms have prompted us to make some improvements to the paper.
>
> ### 2.1. Use of the min operator between two terms in Eq.8:
> Thank you for highlighting this issue. We require that the $P_{\phi} (G | \mathbf{x_i})$ sum to one. This is important because it means that our loss function in Eq. (6) is a correct approximation of the expectation in Eq. (5). Moreover, during inference, we use Eq.(4) to determine $ P(E^l=1 | \mathbf{x_i} ) $ at a gate, and this calculation can lead to probabilities greater than 1 if $P_{\phi}(G | \mathbf{x_i})$ does not represent a correctly normalized probability distribution. The question is then how to map from the unnormalized $\{g^l_{\phi}\}$ provided as output from the neural networks to normalized $P(G|\mathbf{x}_i)$. An immediate thought would be to divide by the sum. But unfortunately, during inference, we are evaluating $g^l$ successively and we do not have access to the sum when we decide to exit. Therefore, we employ the $\min$ operator to ensure that the sum of  $P(G^l|\mathbf{x}_i)$ never exceeds $1$. We have added some text around equation (8) to explain this design choice. We could consider other forms of normalization, provided they can be implemented sequentially, i.e., we need to be able to perform the proposed normalization at layer $l$ without the knowledge of any $g^{l'}(\mathbf{x}_i)$ values for $l'>l$. We would be happy to conduct and include an ablation study if there is an approach that seems a natural alternative to you, but we are not sure what the obvious alternative would be.
>
> ### 2.2 Evaluation only on frozen-backbones:
> This is a valid point. We intentionally focused on fixed backbones because our goal was to develop a mechanism suitable even for very large models where even fine-tuning is a major burden. But we have conducted some additional experiments to explore this direction. In the Observation 5 in the revised paper, we include results where we combine the Dynamic Perceiver with JEI-DNN (our proposed method). We use the Dynamic Perceiver framework to jointly train the backbone and the IMs and then use JEI-DNN to jointly train the gates and the IMS. In our current experiment, this is not true end-to-end joint learning, because we do not jointly train IMs, gates, and backbone, but the combined process does train all three components. Figure 6 shows that using JEI-DNN achieves an accuracy improvement of approximately 0.5-1\% compared to Dynamic Perceiver on the CIFAR100 dataset when using MobileNetV3 as a backbone, when using 50-60 percent of the mul-add operations that are used in the original architecture. We are conducting similar experiments on ImageNet and hope to report the results before the end of the response period. We are also exploring performance for true end-to-end training when we allow the backbone model parameters to update as well as the IMs and gates. Here we retain our JEI-DNN bi-level optimization learning procedure, but update the backbone model parameters at the same time as the IM parameters.
>
> ### 2.3. ImageNet experiments:
> We have commenced experiments on ImageNet and obtained preliminary results. The experiments are ongoing, and we will finalize and report the results at least one day before the end of the response period. Although our introduced gates and IMs are lightweight and have few parameters, training with ImageNet is time-consuming, even with relatively powerful servers.
>
> ### Minor comments:
> That was a mistake, thank you for pointing it out.

---

> > ### Comment · Reviewer_CAtU · 2023-11-22
> >
> > Dear authors,
> > Thank you very much for your commitment to address all raised comments, and the significant work it required to conduct all the additional experiments included in the rebuttal.
> >
> > At this stage, all of my raised concerns have been satisfactorily addressed. Additionally, I believe that the additional results targeting ImageNet as well as the application of the proposed approach to Dynamic Perceiver render the effectiveness of the JEI-DNN notably more convincing.
> >
> > As such, I am increasing my score to Accept.

---

### Official Review · Reviewer_KZAV · 2023-10-31

**Soundness:** 2 fair
**Presentation:** 2 fair
**Contribution:** 3 good
**Rating:** 6
**Confidence:** 4

**Summary:**

This paper presents an early-exit dynamic neural network architecture, JEI-DNN, that augments a backbone with lightweight, trainable gates and intermediate classifier that are jointly optimized. The gates are trained through a surrogate binary classification task, focusing on the optimization for assigning the most cost-effective early-exit classifier for each input. The results are presented on the cifar and svhn datasets for the T2T-ViT backbone.

**Strengths:**

- Analysing the conformal intervals for the early exiting classifiers is interesting and it's promising to see the proposed method yields better uncertainty characterisation.
- The ablation study in 8.6 clearly shows the merits of learnable GMs versus other design choices.
- Figure 2 a) and b) very clearly show the early exiting patterns for the method and competing approaches.

**Weaknesses:**

- It is unclear what the major contribution of the paper is. The use of learned early exiting gates and intermediate classifiers, trained jointly with the backbone is common practice for many early exiting architectures. After all the approximations described in 4.1, the loss term for the gating modules turnes into the common practice of summing the losses of independent binary early-exiting classifiers. The main task loss is similarly aligned with prior work from Han et al. (2022b).

- Constructing surrogate binary targets for the learned gates is also common, e.g. FrameExit by Ghodrati et al. CVPR 2021.

- Figure 2a) The fact that samples do not exit at all from the first few early exiting layers in JEI-DNN is puzzling and I am wondering if it is due to the choice of specific hyperparameters that prevent early exiting from these layers. As can be seen, the accuracy of the IM at layer 5 & 6 is far higher that the overall performance. It is conceivable that a good accuracy higher than the green dashed line is still achievable by the earlier classifiers at least for a proportion of samples. Is there any hyperparamer that could potentially give more control into the exiting pattens of JEI-DNN?


- All the results in the paper are limited to the T2T-ViT backbone. The authors should show the efficacy of their method for a larger variety of backbones, preferably to models that are established in the early exiting literature such as MSDNet, DenseNet, etc.

- Comparison of the accuracies among T2T-Vit and MSDNet architectures in Fig 5 seem unfair. Most of the gain in accuracy comes because of the more powerful transformer-based backbone and not because of the efficacy of the early exiting approaches. In fact, MSDNet and RANet show more robust early exiting results compared to the results reported for T2T. E.g. MSDNet retains the original accuracy of the model after almost 50% compression. The performance of the proposed JEI-DNN approach in comparison drops very rapidly even with 25% compression.

- The method is only evaluated on three small-scale datasets: CIFAR10 & 100, SVHN. The authors should consider expanding the evaluation to ImageNet.

**Questions:**

- The assumption that a sample exited at a gate at layer $l$ should also exit from any late stage gate seems against the prevalent view of $overthinking$:
"Overthinking is computationally wasteful, and it can also be destructive when, by the final layer, a correct prediction changes into a misclassification." by Kaya et al. ICML 2019.
How did you make this assumption?

---

> ### Author Response · Authors · 2023-11-19
> **Response 1/4**
>
> Thank you for the detailed and thoughtful review. You raised some excellent points that made us think carefully, leading to modifications and experiments that have provided insights and improved the paper.
>
> ### 1.1 Contributions:
> The major contributions of the paper are as follows. (1) We propose a novel, robust approach to jointly train the inference modules and decision gates in early-exit neural networks. As explained below, we disagree with the reviewer that this is common practice. (2) We introduce a novel modeling of the probability of exiting, representing it as a constrained sum, rather than a product. This leads to considerably more robust learning performance. (3) We provide a bi-level optimization formulation of the learning task. By decomposing the problem this way, we further encourage stable and efficient training. (4) We demonstrate that our approach results in much better calibration leading to significantly improved conformal intervals.
>
> ### 1.2 Prior Work:
> We disagree that the use of learned early exiting gates and intermediate classifiers, trained jointly with the backbone is a common practice for early exiting architectures. If the reviewer could provide a reference, we would be very happy to discuss more concretely and compare with our approach. We compiled an extensive table (Table 4 in the Appendix) of the existing literature with this specific categorization (non-fixed backbone) to highlight this.
>
> For those methods that train their proposed backbone architecture and the IMs jointly (non-fixed backbone), the architectures are specifically designed for early-exit. Even the more flexible models, such as the Dynamic Perceiver (Han et al., 2023) are restricted in their application; for example the Dynamic Perceiver, in its proposed form, can only be used with CNNs. In contrast, our approach can be easily applied to a transformer, a CNN, or a graph neural network. We demonstrate this better in supplementary experiments that we have done at the request of reviewers. But beyond this, to the best of our knowledge, all of the state-of-the-art methods that train the backbone use fixed-threshold gating mechanisms. For instance, the SOTA early-exit models (e.g. Dynamic Perceiver - Han et al. (2023)), do not train gates.
>
> For the fixed backbone case, which is the setting of our work, the most common approach is to either focus (i) on training the IMs using fixed gates; or (ii) on training only the GMs with a fixed backbone and pre-trained IMs. For example, the method proposed by Han et al. (2022b), L2W, does use a trainable gating mechanism. It trains the inference modules. It is this specific difference in training approach that leads to a major difference in exit behaviour, which the reviewer correctly pointed out in Figure 2.
>
> Another reviewer kindly pointed us to a work - Scardapane et al. (2019) -that does perform joint training of the gates and the IMs (with a fixed backbone). We have added it in our Table 4, and provided results comparing to it in Section 8.13, showing that our JEI-DNN significantly outperforms.
> Compared to the proposal in Scardapane et al. (2019), a key innovation of our work is (i) a novel modelling the exit probabilities; and (ii) the bi-level optimization strategy. These combine to make the learning procedure much more robust.

---

> ### Author Response · Authors · 2023-11-19
> **Response 2/4**
>
> ### 1.3. Similarity of Loss Terms to Prior Work:
>  The critical aspect of our loss formulation and bi-level optimization is the interaction of the learning for the gates and IMs. While there is a superficial similarity between the forms of the losses and the losses in previous work, the model parameterizations and learning process are considerably different. Specifically, it is the combination of the losses that is important. The two loss terms interact via the bi-level optimization; updates of the IM parameterss continually influences the target of the binary tasks of the GMs, and updates of the GM parameters change the weights in our weighted cross entropy loss.
>
> Starting with the gate loss, while it is true that we sum the losses of independent binary variables $g^l_{\phi}$, our binary variables $g^l_{\phi}$ do not directly correspond to exit probability in the way that is common in the literature. In particular, we do not set $ P(G=l)$ $= g^l_{\phi} \prod_{i=1} ^{l-1} (1-g^i_{\phi}) $, as most learnable GMs approaches do. The parameterization of our exit probabilities given by the equations (7) and (8) (and therefore our whole gating mechanism) is different and novel.  We acknowledge that the way we described our gate mechanism might have been too compact, so we have provided an additional figure to clarify our architecture.
>
> Next, we turn to the IMs and our main task loss. By dividing the parameters in our bi-level approach, we obtain two losses, one for the gate parameters $\phi$ and one for the IMs parameters $\theta$. The loss for the IM parameters includes a term with a weighted cross entropy loss, but, critically, the weights are driven by the probability of a sample exiting at that given gate ($P_{\phi}(G=l|\mathbf{x}_i)$). In contrast, the weights in the formulation of Han et al. (2022b) are not connected to the exit probability.
>
> We arrive at the IM loss by starting from a principled combined loss that encompasses both the GMs and the IMs, balancing accuracy and computation. In contrast, Han et al. directly start from the weighted IM loss formulation, with the weights being modeled by a weight predictor network (WPN). There is no interaction with the IM parameters during training. The WPN takes as input the losses at each IM to predict the weight, so it cannot be used at inference. As a result, whereas our weights are tied to the probability of exiting, as they should be, the weights in the loss of Han et al. (2022b) are heuristic, selected somewhat arbitrarily to reduce the train-test mismatch. Our loss incorporates the gate parameters that are actually used at inference, while Han et al. (2022b)'s loss does not.
>
> All of these differences are major. We can confirm empirically that these differences are major because they lead to significantly different behavior, as shown in Figure 2.
>
> ### 1.4. Constructing surrogate problems:
> We agree that constructing a surrogate problem is not novel and is not part of our claimed contribution (it does not appear in the paragraph at the end of the introduction). It is merely a detail of the optimization process.
>
> ### 1.5 Figure 2a exit behaviour:
> This is a very interesting point and we appreciate that you have raised it. When examining the accuracies, we need to be careful because the accuracies at layers 5 \& 6 in Figure 2a) are only *for the samples that exit there*. These accuracies are very high, showing that the gates have effectively learned to exit the correct samples.  The accuracy at layers 5 \& 6 for all the samples of the dataset is much lower, as we can see in Figure 2b). This figure is for a specific $\lambda=1$ (the corresponding operating point is marked with a star in the left panel of Figure 1). For that particular parameter, it is not unexpected that the gates have learned to avoid the first 3 gates entirely and only start to exit a very few samples at gate 4.
> The decision being made is whether the drop in cross-entropy loss (in going to gate 5 from gate 4, for example) is worth the additional cost of inference. For the choice $\lambda=1$, it is not the case for many samples. While IM 4 may still be able to classify some subset of the points with accuracy of 90 percent, IM 5 would classify that same subgroup with accuracy 95 percent. In Appendix 8.15. of the revised paper we now provide the same figure for a higher $\lambda=3$,  corresponding to a greater penalty on computation, and $\lambda=0.5$, corresponding to a reduced penalty on computation, with more focus on accuracy. For the higher $\lambda=3$, we observe that now the gates decide to exit many more samples at earlier gates (2,3,4).  This behavior is a consequence of our novel early-exit framework. These results indicate that the $\lambda$ parameter does provide meaningful control over where samples exit.

---

> > ### Author Response · Authors · 2023-11-19
> > **Response 3/4**
> >
> > ### 1.6. Limitation to the T2T-ViT backbone:
> >  Thank you for the suggestion to conduct experiments with different backbones. During the response period, we have conducted experiments using the following two backbones: (i) Dynamic Perceiver on MobileNet-v3; (ii) GRIT - a graph transformer. Dynamic Perceiver is an early-exit architectural extension for CNNs that is shown to outperform MSDNet and DenseNet, so we consider that using it as a backbone is sufficient to demonstrate that the proposed JEI-DNN can be combined with state-of-the-art early-exit architectures and CNNs. The successful application of our method to a graph transformer illustrates the flexibility of the proposed approach. The results for these two architectural settings are reported in Appendix Sections 8.12 and Observation 5 of the revised paper.
> >
> > ### 1.7. Figure 5 - Unfair accuracy comparison between T2T-Vit and MSDNet architectures:
> >  We recognize that the comparison between the T2T-Vit and MSDNet architectures is unfair. The main point of the comparison was to demonstrate that the proposed JEI-DNN learning procedure can achieve performance improvement because it is sufficiently flexible that it can be incorporated with state-of-the-art architectures. The text in the original version of the paper was  insufficiently clear on this point. In contrast, MSDNet  and RANet are dedicated architectures that incorporate design decisions like skip connections and multi-scale density that are specifically tailored for early-exiting. We thus cannot use MSDNet in conjunction with a state-of-the-art model to improve performance. The two SotA baselines used in the original version of the paper (BoostNet and L2W-DEN) provide architecture-agnostic training for early-exiting that aims to improve the accuracy-cost trade-off. As such, a fairer and better comparison is to compare these two training procedures with ours on a given architecture as we did throughout most of the paper using T2T-ViT.  Another reviewer raised the point that the Dynamic Perceiver (Han et al., 2023) can probably be considered the state-of-the-art for dynamic early-exit architectures. The results presented by Han et al. show clear outperformance of MSDNet and RANet. We have now conducted experimental comparisons with Dynamic Perceiver, which can augment almost any CNN model to perform efficient early-exiting. The results show that our learning procedure improves upon the performance of Dynamic Perceiver, so we consider this to be a much fairer and more meaningful comparison with an existing, state-of-the-art dynamic early-exit architecture.
> >
> > In the current version of the paper, we have moved the comparison with MSDNet, RANet, BoostedNet and L2W-Den to an Appendix and included a much clearer statement that the results should be interpreted with care because the performance discrepancy is largely due to the backbone. We would also be happy to remove this comparison from the paper entirely, since we consider the results to be somewhat redundant in light of the new dynamic perceiver results.
> >
> > The new experiments demonstrate that JEI-DNN can be used to improve the performance of Dynamic Perceiver by adding small trainable gates to the architecture, and then jointly learning the IM and gate weights. Currently, we include the results for CIFAR100 in ``Observation 5'' in the results section, but we are compiling ImageNet results and hope they will be finalized before the end of the response period.
> >
> > ### 1.8. Larger scale experiments:
> > We have commenced experiments on ImageNet and obtained preliminary results. The experiments are ongoing, and we will finalize and report the results at least one day before the end of the response period.

---

> > > ### Author Response · Authors · 2023-11-19
> > > **Response 4/4**
> > >
> > > ### 1.9 Assumption that a sample exited at a gate at layer should also exit from any late stage gate vs. Overthinking:
> > >  Overall, the entirety of our proposed approach is very much aligned with the quoted sentence by Kaya et al. (2019). We aim to train to exit a sample at the earliest possible layer where the probability of correct classification is sufficiently high. Having said that, we understand that you are referring to our training assumption for optimizing the gates in Section 5, where we state that ``we assume that it should also exit at any later gate''. Kaya et al. 2019 study shallow-deep networks, where intermediate backbone layers are trained to provide outputs that can be used effectively for classification. In contrast, the backbones we employ are not shallow-deep. For a shallow-deep architecture, it is much more likely that the IM based on layer 5, for example, might produce a correct answer, whereas the IM at layer 9 might not. In our backbones, where the intermediate features are not intended to be used for immediate classification without processing by subsequent layers, this phenomenon is much less common. To quantify this, we calculated a conditional accuracy for each IM, where the conditioning is on the previous IM correctly classifying the sample. The results are included in Figure 20 in Appendix 8.16 in the revised version of the paper. The figure shows that the conditional accuracy exceeds 90 percent for all IMs. This compares to unconditional accuracies of 40 and 65 for IMs 2 and 3. This observation supports the intuition behind the assumption we made, namely that once one IM manages to make a correct classification, all subsequent IMs are likely to do so and we should exit as soon as possible (set all the targets to 1).
> > >
> > > Having conducted this test, your insightful observation prompted us to consider the necessity of the assumption. Indeed, there is no reason that JEI-DNN cannot be used in conjunction with a shallow-deep network, and there these assumptions may be flawed. For this reason, we identified an alternative approach of setting a later gate's binary target to 1 only if its combined accuracy and inference cost (for the current IM parameters) is less than all subsequent gates. With this strategy, we avoid the assumption that all subsequent IMs will be correct. We have conducted preliminary experiments using this approach; for the t2t-vit-7 backbone on the SVHN dataset, the performance is almost the same as the original scheme.

---

> > > > ### Comment · Reviewer_KZAV · 2023-11-22
> > > >
> > > > I would like to thank the authors for their detailed responses and the newly added experiments that resolve most of my concerns (including adding new backbones and experiments on ImageNet). I raise the contribution score to 3, and the overall score to 6, accordingly.

---

### Author Response · Authors · 2023-11-19
**General response**

We would like to express our gratitude to the reviewers for their detailed and insightful reviews. They have prompted some careful thought about aspects of our paper and led to additional experiments to further explore the behaviour and confirm performance on a larger dataset. We appreciate that all reviewers acknowledged the strengths of our work. In particular, reviewers indicated that our work addresses an interesting problem [R2-R4], is well-motivated and provides a clear problem formulation [R2]. Additionally, the
reviewers acknowledged a technically sound methodology with well-motivated design choices [R2,R3]. In general, all reviewers were satisfied that the experiments demonstrated the effectiveness of the method and especially the discussion of the findings was also considered thorough [R2].

In addition to their recognition of these strengths of the paper, the reviewers provided important constructive criticism.

[C1] Importantly, all reviewers expressed that the experimental results would be more compelling if we analyzed a larger dataset, and in particular ImageNet. [C2] There were also requests to examine performance for additional backbone architectures, and in particular the state-of-the-art dynamic exit architectures. Reviewers also asked whether it was possible to generalize the work to fine-tune the backbone while learning parameters. [C3] Several reviewers observed that our contributions and methodology could be better highlighted and explained, particularly with the introduction of additional figures that depict the methodological approach.

We have addressed the common concerns as follows:

- [CR1] ImageNet: We have commenced experiments on ImageNet and obtained preliminary results. The experiments are ongoing, and we will finalize and report the results at least one day before the end of the response period. Although our introduced gates and IMs are lightweight and have few parameters, training with ImageNet is time-consuming, even with relatively powerful servers. We are also working on demonstrating how the proposed learning methodology (JEI-DNN) can be combined with the Dynamic Perceiver (a state-of-the-art dynamic exit architecture) to achieve an even better trade-off between accuracy and computation.

- [CR2] Additional Backbone Architectures and Dynamic-exit backbone*: During the response period, we have conducted experiments using the following two backbones: (i) Dynamic Perceiver on MobileNet-v3; (ii) GRIT (Ma et al, 2023) - a graph transformer. Dynamic Perceiver is an early-exit architectural extension for CNNs that is shown to outperform MSDNet and DenseNet, so we consider that using it as a backbone is sufficient to demonstrate that the proposed JEI-DNN can be combined with state-of-the-art early-exit architectures and CNNs. The successful application of our method to a graph transformer illustrates the flexibility of the proposed approach. The results for these two architectural settings are reported in the revised version of the paper in Appendices 8.10 and 8.11. To summarize, we demonstrate that we can obtain a 0.5-1\% accuracy improvement compared to the Dynamic Perceiver for the operating region where we use 40-60\% of the mul-adds of the original architecture (for the CIFAR100 dataset). We hope to report the results of a similar experiment for ImageNet prior to the end of the response period. For the graph transformer GRIT, we demonstrate that we can consistently improve relative to Boostnet, one of the few baseline methods that can also be applied to general architectures. For a super-resolution graph dataset, JEI-DNN can maintain an accuracy above 90 percent (reduction by 5\% relative to performance of original architecture) while reducing mul-adds by 30 percent.

- [CR3] Methodology figures and explanation: We have made small, but important, modifications to the text of the introduction and methodology to better highlight our contributions and make the presentation of the methodology clearer. The most important modification is the introduction of a figure, which should provide readers with a better understanding of the joint learning strategy and the employment of bi-level optimization.

Other questions and concerns arose in individual reviews and we discuss them and explain how we have addressed them in individual responses to the reviewers below.

---

> ### Author Response · Authors · 2023-11-22
> **Additional experiments on more datasets and architectures**
>
> Our most recent revised version contains the following:
> * As promised, we have added results on ImageNet in Observation 5 of our main paper and in Appendix 8.13. The ImageNet experiments show JEI-DNN on two types of backbone architectures: a general architecture (T2T-ViT-14) and a state-of-the-art early-exit dedicated architecture (Dynamic Perceiver).
> * For the general architecture (T2T-ViT-14), we compare to the best-performing training baseline, L2W-DEN. For equivalent accuracy, we can reduce the inference cost by 1-2 percent over an important operating regime. The results on ImageNet are thus in line with the results of our other experiments on the other datasets.
> * For the early-exit specialized architecture (Dynamic Perceiver), which already achieves an impressive trade-off between inference cost and accuracy, we show that employing JEI-DNN can offer further improvement, reducing the inference cost by an additional 1-2 percent.
> * Moreover, we additionally presented an experiment for end-to-end training where we combined the training of our JEI-DNN method with the Dynamic Perceiver in Appendix 8.17. Fully end-to-end training (JEI-DNN+backbone) offers the best performance for the CIFAR100 dataset.

---

### Meta-Review · Area_Chair_UJdG · 2023-12-06

**Metareview:**

The paper proposes JEI-DNN, a method for converting an existing classifier into an early-exit procedure for improved efficiency. Crucially, due to probabilistic gates, the training and evaluation settings are not mismatched. The reviewers found the paper interesting and the empirical evaluation to be convincing. While the reviewers initially raised several concerns about weaknesses, e.g., a lack of ImageNet experiments, these concerns were largely addressed by the authors.

**Justification For Why Not Higher Score:**

While the paper presents an interesting idea and useful results, I am unsure to what extent the idea will be applicable to a broad audience. While such ideas could be applied broadly, the setting strikes me as somewhat niche.

**Justification For Why Not Lower Score:**

The paper provides a novel approach for early-exit networks with a clear presentation and empirical evaluation. The reviewers are in agreement that the paper warrants publication.

---

### Decision · Program_Chairs · 2024-01-16

Accept (poster)